# Sensitivity of fatigue reliability in wind turbines: effects of design turbulence and the Wöhler exponent

Shadan Mozafari[1,*], Paul Veers[2], Jennifer Marie Rinker[1], and Katherine Dykes[1]

[1]Department of Wind Energy, Technical University of Denmark, Roskilde, Denmark
[2]National Renewable Energy Laboratory (NREL), Golden, CO, USA.

**Correspondence:** Shadan Mozafari (Shad.mzf@gmail.com)

**Abstract.**

Fatigue assessment of wind turbines involves three main sources of uncertainty: material resistance, load, and the damage accumulation model. Many studies focus on increasing the accuracy of fatigue load assessment for improving the fatigue reliability. Probabilistic modeling of the wind's turbulence standard deviation is an example approach for this purpose.

Editions 3 and 4 of the IEC standard for the design of wind energy generation systems (IEC 61400-1) suggest different probability distributions as alternatives for the representative turbulence in the normal turbulence model (NTM) of Edition 1. There are debates on whether the suggested distributions provide conservative reliability levels, as the established design safety factors are calibrated based on the representative turbulence approach. The current study addresses the debate by comparing annual reliability based on different scenarios of NTM using a probabilistic approach. More importantly, it elaborates on the relative importance of load assessment accuracy in defining the fatigue reliability.

Using the DTU 10-MW reference wind turbine and the first-order reliability method (FORM), we study the changes in the annual reliability level and its sensitivity to the three main random inputs. We perform the study considering the blade root flapwise and the tower base fore-aft moments assuming different fatigue exponents in each load channel.

The results show that integration over distributions of turbulence in each mean wind speed results in less conservative annual reliability levels than representative turbulence. The difference in the reliability levels varies according to turbulence distribution and the fatigue exponent. In the case of the tower base, the difference in the annual reliability index after 20 years can be up to $50\%$. However, the model and material uncertainty have much higher effects on the reliability levels compared to load uncertainty. Knowledge about such differences in the reliability levels due to the choice of turbulence distribution is especially important, as it impacts the extent of lifetime extension through reliability reassessments.

**keywords**: Wind turbine reliability, Fatigue reliability, Uncertainty quantification, Sensitivity analysis, Normal Turbulence Model, Turbulence distribution

## 1 Introduction

Fatigue reliability of a structure is its ability to withstand cyclic loading during the design life. Fatigue life is a highly sensitive and uncertain variable (Veers, 1996). In the case of wind turbines, the random and variable amplitude loading and the complex-

ity of the structural system increase such uncertainty. In addition, there is a high level of uncertainty in material strength and in the simplified models commonly used for counting cycles or describing the material properties and damage accumulation. The probabilistic approach for fatigue reliability assessment involves probabilistic modeling of the crucial random inputs leading to a more robust analysis and design of the wind turbines against fatigue (Choi et al., 2007). The IEC design standard introduces a semi-deterministic approach that includes safety factors to account for the uncertainty in different inputs. These factors are

calibrated based on probabilistic reliability assessment aiming at an acceptable target reliability level at the end of design life. Treatment of the uncertainty of the different random inputs is an important part of the probabilistic reliability assessment.

Stensgaard et al. (2016a) show that wind climate parameters contribute to about 10%–30% of the total uncertainty in the reliability estimation. Furthermore, the sensitivity analysis results in (Stensgaard et al., 2016a; Dimitrov et al., 2018; Robertson et al., 2019; Murcia et al., 2018) reveal that after the mean wind speed, the standard deviation of the wind (turbulence) has the

largest impact on the equivalent fatigue load levels in most of the load channels. Thus, accurate modeling of the turbulence at the design level is crucial for decreasing the uncertainties in the fatigue loads.

The IEC standard (IEC 61400-1, 2005) suggests the Normal Turbulence Model (NTM). The model is mainly based on a representative value in each wind speed bin for estimating turbulence. The representative value is the 90% quantile of a lognormal

distribution. However, the standard also allows using the whole lognormal distribution instead of a single value. Edition 4 of the IEC standard (IEC 61400-1, 2019) suggests a Weibull distribution with the same 90% quantile magnitude.

Some studies investigate the performance of different NTM approaches in describing the site turbulence conditions. For example, Ren et al. (2018) investigated the suitability of the 90% quantile recommendation for describing onshore conditions

and showed that it overestimates the turbulence. Thus, they proposed a three-parameter power-law model for turbulence intensity. In addition, Øistad (2015) investigated the performance of representative turbulence recommendations on a case with both onshore and offshore wind conditions and concluded that the model overestimates the turbulence levels in both cases. Other similar studies focused on offshore conditions (Wang et al., 2014; Türk and Emeis, 2010; Tsugawa et al., 2015) show the inaccuracy of the IEC standard recommendations in representing the turbulence in real wind fields. In addition, Ishihara (2012)

presented the same results by studying the compliance of the lognormal distribution of the IEC standard with measurements in an offshore wind farm. As another example, Dimitrov et al. (2017) showed that for a case study site, the lognormal distribution provides overconservative turbulence expectations, and thus, it suggests a 2-parameter Weibull distribution. Emies (2014) showed an offshore case within which the IEC Normal Turbulence Model provides underconservative results for turbulence intensity in low mean wind speeds and over-conservative values in higher mean wind speeds. A few studies like (Larsen, 2001;

Wang et al., 2014) explicitly introduced other approaches for modeling offshore turbulence, as they proposed that the NTM is overconservative for all offshore cases.

In addition, there are some studies on the loads corresponding to each turbulence characterization approach. The results of these studies vary from each other. For example, some pieces of research (Ernst et al., 2012; Dimitrov et al., 2017; Hansen and

Larsen, 2005) showed that the IEC representative and full lognormal distribution models of turbulence are very conservative in the case of blade loads and proposed new, less conservative turbulence models. The results of Søndergaard and Jóhannsson (2016) also conclude that the lognormal assumption is conservative. They suggested the Weibull distribution as a less conservative choice in terms of resulting blade fatigue and extreme loads. On the other hand, the results of (Stensgaard et al., 2016a) show that following the suggested 90% quantile level for turbulence in the IEC standard leads to an accurate assessment of the blade root flapwise bending moment while producing a conservative assessment of the tower bottom fore-aft bending moment and low-speed shaft torque. It has to be noted that these comparisons are made with real data in which the wind shear is also variable (and a function of turbulence), and affects the special load channels.

As a second matter, the effect of uncertainty in the material properties on fatigue reliability assessment is also covered in many studies. According to previous research (Veers, 1996; Zaccone, 2001), the uncertainty inherent in the material properties, including physical, modeling, and measurement uncertainty, represents almost half the total uncertainty in the fatigue damage. The results of the sensitivity analysis in (Velarde et al., 2020)) and Ronold et al. (1999) showed that the uncertainties related to the material resistance model have the greatest influence on fatigue reliability. Bacharoudis et al. (2015) presented the high sensitivity of wind turbine blade reliability to the measurement uncertainty in the material properties of the composites using the DTU 10-MW wind turbine as the case study.

All in all, there are many studies on the accuracy and performance of the representative turbulence, and they all show the need for transition from such a model to lognormal and Weibull distributions, especially in the case of offshore wind farms with overall lower turbulence levels. However, they have not compared different scenarios to each other in general design conditions. It is still debated whether the two distributions always provide lower reliability levels for different load channels compared to the representative value approach. The current work addresses this gap and such debate. Knowing the difference between the reliability levels when following different NTM approaches is especially important because the established safety factors for the semi-deterministic approach in the IEC design standard are calibrated based on the representative turbulence approach. Thus, if reliability levels are assessed using the same safety factors while characterizing the standard deviation turbulence by distribution, the semi-deterministic approach and probabilistic approach do not meet at the same reliability level at the end of the design life. It is important to ensure the two alternative distributions are never underconservative. Furthermore, knowing the differences in reliability levels can be an asset for fast initial estimation of the possible extension of a lifetime when the considered approach in the design phase is known. Such information about the design assumptions is also crucial for more accurate lifetime extension assessments.

In addition, considering the results of the previous studies on the importance of material uncertainty in fatigue reliability, it is valuable to investigate how all the efforts for accurate modeling of the turbulence will transfer to a more robust reliability assessment. In the present study, we reveal how different approaches in the IEC standard for a general case can change the distribution of the fatigue loads. We also study the sensitivity of the reliability to the change in the fatigue load compared to its

sensitivity to variations in other random inputs.

The results of the current study cover blade flapwise and tower base fore-aft load channels in a large wind turbine (10-MW) from IEC class 1A. We study the difference in distributions of the damage equivalent load (DEL) considering different NTM representations using many aeroelastic simulations and bootstrapping technique. In addition, the results show the overall
importance of DEL variation due to turbulence model by revealing the relative effects of load uncertainty on the reliability. Knowing the extent to which the various sources of uncertainty affect reliability can help designers and researchers focus on effective areas to get robust reliability levels. The fatigue exponent is the exponent to which the load is powered in the damage models commonly used (as in the current study). Thus, it directly changes the share of the loads in the fatigue damage, and it also changes the distribution of DEL (Mozafari et al., 2023). We investigate the results of the reliability study in different
fatigue exponent levels in each load channel. An important note is that since the DTU 10-MW turbine is not designed against fatigue, we observed low reliability levels in the blade root and the tower base. Thus, the mean value of the material properties is scaled while keeping the coefficient of variation the same. To lower the errors in the first-order reliability method (FORM) in the case of the blade (high fatigue exponent and thus high nonlinearity), we calibrate the mean material strength to obtain lower probabilities of failure for the sake of accuracy. The reliability levels are therefore not based on real data. In other words,
the trends and effects, which are the purpose of the current study, are reliable, but the reliability levels are not.

We provide information about the wind turbine case study and the the aeroelastic simulations in in Sect. 2.1 and Sect. 2.2, respectively. Sect. 2.3 provides information about the assumptions and the mathematical relations used in the current study for assessing fatigue loads, post-processing simulation results, reliability assessment, and sensitivity analysis methods. Then,
Sect. 3 presents the results in three parts: Sect. 3.1 covers the results of DEL distributions, Sect. 3.2 shows the results of the reliability assessments, and Sect. 3.3 provides the sensitivity analysis of the reliability. Finally, Sect. 4 contains the conclusions of the study together with limitations and suggestions for future research in the area.

## 2  Methodology

We use 10-minute aeroelastic simulations to obtain the load time series and estimate the fatigue reliability based on them. Fig.
1 presents a general, schematic view of the main random inputs of the fatigue reliability assessment and the procedure for the analysis in the current study. The following sections will provide more details.

Sect. 2.1 explains the case study wind turbine specifications, and Sect. 2.2 introduces the properties of the simulations. Finally, Sect.2.3, explains the mathematical relations and methods we use.

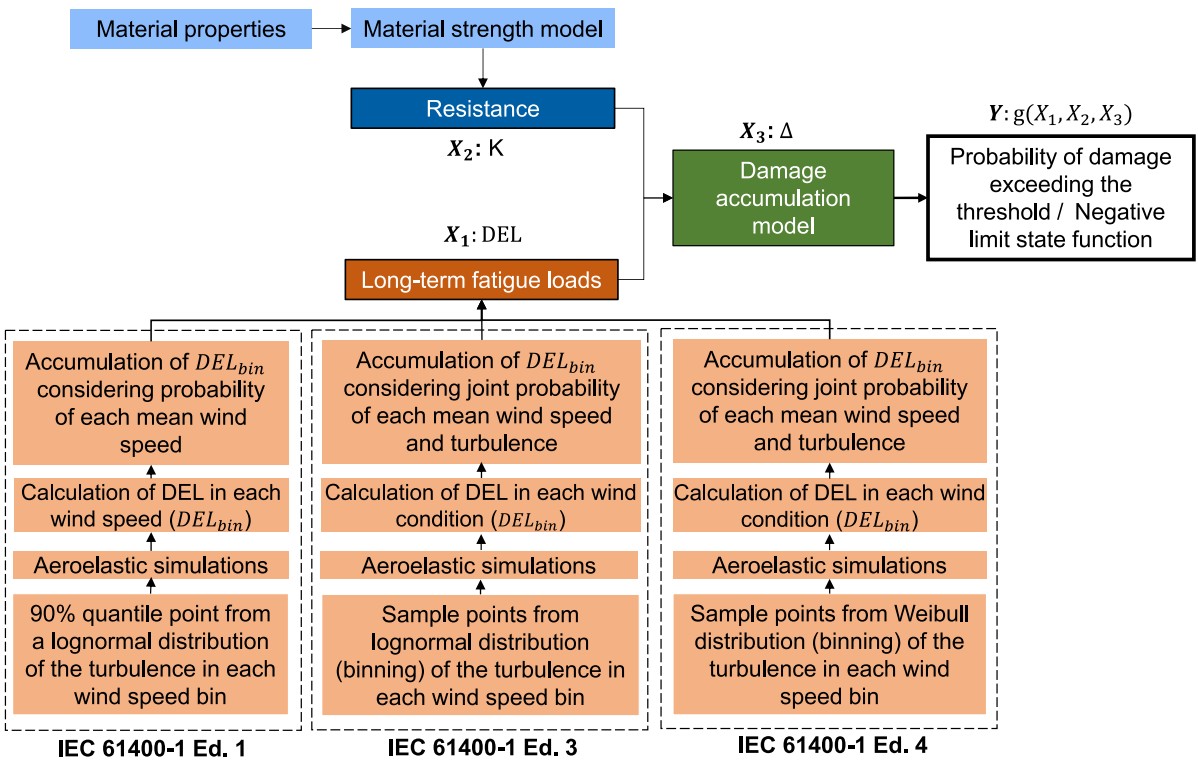

**Figure 1.** Flowchart of the procedure, main random inputs ($X_i$), and the output ($Y$) considered in the present study for the fatigue reliability assessment

## 2.1 The case study wind turbine

Our case study is the DTU 10-MW reference wind turbine (Bak et al., 2013). The DTU 10-MW is from IEC standard class 1A (IEC 61400-1, 2019) with a rotor diameter of 178.3 m and a hub height of 119 m. It is rated at a power of 10 MW and a mean wind speed of 11.4 m/s. The cut-in and cut-out mean wind speeds are 4 m/s and 26 m/s, respectively. The blade in the current case study is made of unidirectional E-glass fiber epoxy, and the tower is made of steel. In the present study, we use the onshore version of the DTU 10-MW wind turbine.

## 2.2 Aeroelastic simulations

We perform three groups of aeroelastic simulations in HAWC2[1] software (Larsen and Hansen, 2007), forming a total of 98,400 10-minute simulations of the DTU 10-MW reference wind turbine. Each simulation group covers an approach for modeling

---

[1]An aeroelastic code for calculating wind turbine response in the time domain – developed in DTU Wind energy department between years 2003-2007

turbulence (standard deviation of wind) in wind speed bins.

The current work only covers normal operating conditions and does not consider the fatigue damage occurring in the fault, idling, start-up or shutdown events. Thus, we perform the simulations based on IEC standard design load case (DLC) 1.2 (IEC 61400-1, 2019) with normal wind condition. As another simplification, we set the wind direction constant equal to zero in all the simulations.

The mean wind speed varies from 4 m/s (cut-in) to 26 m/s (cut-out) in bins of size 2 m/s. Simulations are performed for 700

s, from which the first 100 s is recognized as transient time and are omitted from the results. The transient time is defined by checking the time of stabilization for tower base side-side moments in high mean wind speeds (20–26 m/s), as this load channel is the one with the longest stabilization time.

We use the Mann turbulence model for modeling the wind field (see (Mann, 1998)). The Mann turbulence boxes contain 8,192 evaluation points in the wind direction for higher resolution and 32 points in the other two perpendicular directions. Each wind

bin (a combination of wind speed and turbulence) has 200 turbulence realizations. We use 200 turbulence seeds, as the results of a previous study (Mozafari et al., 2023) show that the estimation of the fatigue loads fairly converges in this number of realizations.

Table 1 presents the specifications of wind load modeling in each group of simulations.

**Table 1.** Specifications of wind modeling in three groups of HAWC2 simulations corresponding to three study cases

| Parameter | Group 1 | Group 2 | Group 3 |
|---|---|---|---|
| Marginal distribution of turbulence | constant | lognormal | Weibull |
| Turbulence levels in each mean wind speed bin | 1 | 20 | 20 |
| Realizations per wind condition | 200 | | |
| Wind shear exponent | 0.2 | | |
| Turbulence model | Mann | | |
| Cut-in mean wind speed (m/s) | 4 | | |
| Cut-out mean wind speed (m/s) | 26 | | |
| Rated wind speed (m/s) | 11.4 | | |
| Size of wind speed bins (m/s) | 2 | | |
| Mean wind speed distribution | Rayleigh | | |
| Yaw angle (degrees) | 0 | | |
| Simulation time (seconds) | 700 | | |
| Initial transient time (seconds) | 100 | | |
| Mann box grids along the wind | 8192 | | |
| Mann box grids in other dimensions | 32 | | |
| Time steps of the simulations (s) | 0.01 | | |

The study considers the flapwise bending moment in the blade and the fore-aft bending moment in the tower base as the main outputs of the simulations and the input for fatigue assessment of the blade and tower.

## 2.3 Mathematical formulations

In the following, we present the mathematical background and relations we use for post-processing simulation load outputs and estimating the corresponding fatigue damage and reliability.

### 2.3.1 Probabilistic modeling of wind

The wind as a random process is mostly described by its mean value and standard deviation (turbulence) at each point of time and space. The IEC standard (IEC 61400-1, 2019) presents a Rayleigh distribution for probabilistic modeling of the mean wind speed at the wind turbine's hub height. Equation (1) presents the cumulative distribution function (CDF) of the mean wind speed based on the suggested Rayleigh distribution.

$$F(V_{hub}) = 1 - e^{(-\pi(\frac{V_{hub}}{2V_{ave}})^2)} \tag{1}$$

In Eq. (1), $F(V_{hub})$ is the CDF. Furthermore, $V_{hub}$ accounts for the mean wind speed at the hub height, and $V_{ave}$ is the annual mean wind speed at the hub height. In the standard wind turbine classes, $V_{ave} = 0.2V_{ref}$ in which the $V_{ref}$ is the 50-year extreme wind speed over 10 minutes. The parameter $V_{ref}$ equals 50 m/s in the IEC class 1 category (IEC 61400-1, 2019), the class of the current case study wind turbine.

The statistical parameters of the wind are correlated. In other words, the standard deviation of the wind (turbulence) changes with a change in the mean level. However, since the IEC design standard suggests binning of the wind speeds (as we do in our simulations), one can use the marginal distribution of turbulence in each wind speed bin. The first option is to consider the constant representative turbulence for each wind speed bin, equal to 90% quantile of the distribution, instead of the marginal distribution. The other option is to consider the whole distribution domain in each wind speed bin. The third edition of the IEC standard (IEC 61400-1, 2005) presents a lognormal distribution as the marginal distribution of turbulence (standard deviation of the wind speed) within each wind speed level. The fourth edition (IEC 61400-1, 2019) suggests the Weibull distribution. Following each distribution, the designer has two options in the IEC standards.

The current study investigates the impact of NTM turbulence characterization choice on design fatigue reliability, as it has not been studied before. For this purpose, we define three cases for the different turbulence characterization approaches: case 1 covers the 90% quantile turbulence value, and cases 2 and 3 refer to the lognormal and Weibull distributions, respectively. Equations (2), (3), and (4) show the CDF, standard deviation, and mean of the suggested lognormal distribution ($T \sim LN(\mu_T, \sigma_T)$), respectively (case 2). In addition, Eq. (5) shows the 90% quantile value of the same distribution (case 1).

$$F(T) = 0.5(1 + erf(\frac{ln(T) - \mu_T}{\sigma_T\sqrt{2}})) \tag{2}$$

$$\sigma_T = \sqrt{ln((\frac{I_{ref}(1.4(\text{m/s}))}{0.75V_{hub} + 3.8(\text{m/s})})^2 + 1)} \tag{3}$$

$$\mu_T = ln(I_{ref}(0.75V_{hub} + 3.8(\text{m/s}))) - \frac{(\sigma_T)^2}{2} \tag{4}$$

$$T_{rep.} = I_{ref}(0.75V_{hub} + 5.6(\text{m/s})) \tag{5}$$

Equations (3) and (4) refer to the NTM model in (IEC 61400-1, 2005). In these equations, $T$ represents turbulence standard deviation, and $I_{ref}$ is the reference turbulence intensity equal to 0.16 for the standard class 1 wind turbines (the current case study). In addition, $\mu_T$ and $\sigma_T$ are the mean and standard deviation of the turbulence as a function of the hub height wind speed ($V_{hub}$). $T_{rep.}$ is the representative turbulence equal to 90% quantile of the lognormal distribution. One must note that Eq. (5) is

not the exact calculation of the representative turbulence and is only a linear regression approximating it.

Equations (6), (7), and (8) present the CDF (considering that the turbulence always has positive values), the shape parameter, and the scale parameters of the Weibull distribution ($T \sim Wbl(K, C)$), which the fourth edition of the IEC standard (IEC 61400-1, 2019) suggests.

$$F(T) = 1 - e^{-(\frac{T}{C})^K} \tag{6}$$

$$K = 0.27V_{hub} + 1.4(\text{m/s}) \tag{7}$$

$$C = I_{ref}(0.75V_{hub} + 3.3(\text{m/s})) \tag{8}$$

In Eqs. (7) and (8), $K$ and $C$ represent the shape and scale parameters of the Weibull distribution, respectively. Weibull is very flexible and can be close to many other distributions, including lognormal, depending on its shape parameter. Figure 2 presents the cumulative probability distributions of cases 2 and 3 in one mean wind speed of 8 m/s. In this plot, the horizontal

axis shows turbulence ($T$) levels and vertical axis refers to $log_{10}(1 - F(T))$, where $T \sim LN(\mu_T, \sigma_T)$ and $T \sim Wbl(K, C)$.

The lognormal distribution is generally heavy-tailed; however, as Fig. 2 shows, for the distribution parameters provided in the standard, NTM with Weibull distribution has a thicker tail. In other words, with the same accumulated probability in the tail (for example, the 10% upper tail), the Weibull tail covers lower turbulence levels than the lognormal distribution and shows

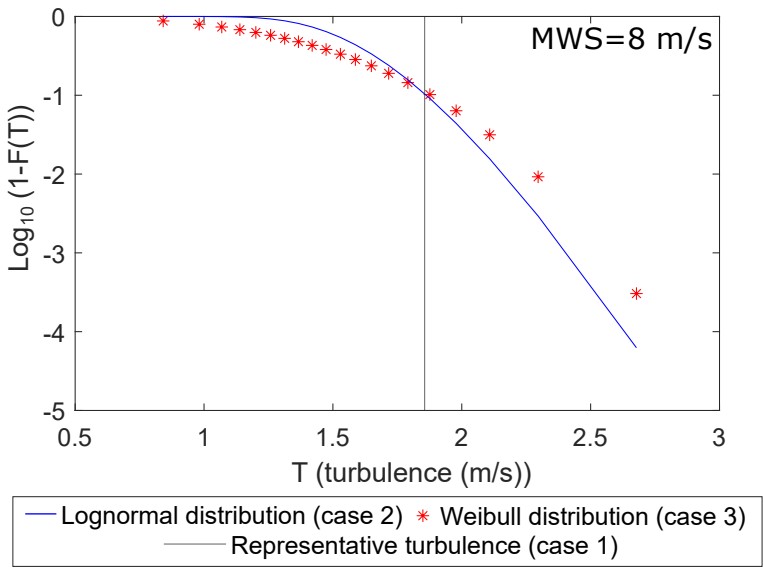

**Figure 2.** Lognormal and Weibull distribution of turbulence at mean wind speed (MWS) of 8 m/s compared to the 90% quantile level (representative turbulence level)

higher standard deviation. Overall, the two distributions have opposite behaviors in the two portions before and after their intersection at 90% quantile.

The Weibull distribution represents higher probabilities in the lower turbulence levels and covers more data from lower values considering the same number of observations. Therefore, we generally expect the use of the Weibull distribution to represent turbulence occurrences to be a less conservative approach. In the following sections, we study this expectation and its effects on the DEL estimations in each case.

### 2.4 Resistance and damage models

Material fatigue resistance tests are often performed under constant-amplitude cyclic loading. The number of cycles that the test specimen can endure in each stress amplitude before failing is collected within S-N curves. Basquin formulation (Basquin, 1910) is a linear regression of the $logN - logS$ data points valid for most materials in the region between approximately $10^3$ and $10^6$ cycles. Equation (9) shows the Basquin relationship.

$$N = kS^{-m} \tag{9}$$

In Eq. (9), $N$ is the number of endured (allowed) cycles in the stress amplitude equal to $S$. Both $m$ (the fatigue exponent, also known as Wöhler's exponent) and $k$ (the Basquin coefficient) in Eq. (9) are material-specific. The Basquin regression model is uncertain. In addition, there is also physical and measurement uncertainty included in the material properties and the fatigue

tests. In the Basquin representation, $m$ and $k$ reflect the same sources of uncertainty because they are strongly correlated (an increase of one decreases the other in the regression). Thus, it is enough to model only one of them as the random variable to describe the scatter of the fatigue strength test data (Kececioglu, 2002; Veers, 1996). Following the mentioned common approach, we only consider the variability in the $k$ parameter in the $logN - logS$ curve and assume Wöhler's exponent to be deterministic. Considering available data regarding average values of $m$, we assume the constant fatigue exponent to be equal

to 10 in the composite case and equal to 3 in the case of steel components. Thus, we consider two values of $m = 10$ and $m = 3$ corresponding to the blade and the tower, respectively. We consider the compression-compression fatigue data (load ratio of R = 10) for the fatigue analysis of the flapwise bending moments according to Mikkelsen (2020) and the resulting time series of the current study.

For variable loading, as in the case of wind turbines, one should use a suitable model to relate the constant amplitude data

to the accumulated damage. Palmgren–Miner (Miner's) rule (Palmgren, 1924; Miner, 1945) is a common linear model for this purpose. Equation (10) presents Miner's rule.

$$D = \sum_{i=1}^{Ns} \frac{n_i}{N_i} \tag{10}$$

In Eq. (10), parameter $D$ accounts for the magnitude of fatigue damage, $n_i$ is the number of cycles of the $i$th load amplitude

in the loading, and $N_i$ is the allowed number of cycles according to the S-N curve of the material. In addition, $Ns$ accounts for the total stress amplitude levels. According to the Miner's rule, failure happens when the summation in Eq. (10) is higher than the material limit, often assumed equal to unity.

The blade's root cross section in the case study wind turbine is nearly circular, and the current study assumes the same. In addition, the pitch angle in the blade root is zero. Thus, the moments along the $x$ and $y$ axes refer to flapwise and edgewise

moments, respectively. In both cross sections of the blade root and the tower base, Eq. (11) can represent the relation between the moments and stresses.

$$S_i = \frac{M_{x_i} c}{I_y} \tag{11}$$

In Eq. (11), $M_{x_i}$ is the moment corresponding to the stress level $S_i$. In the present study, the direction $y$ corresponds to the global direction of the wind in the HAWC2 simulations since we are considering the root in the blade, and the local coordinate

system of the tower base section is also aligned with the global coordinate system. Thus, we consider moments in the perpendicular direction ($M_x$). In addition, the section parameters $c$ and $I_y$ are the radius and the moment of inertia in the direction of the wind (perpendicular to $M_x$). Table 2 presents the values for $c$ and $I_y$ parameters in the blade's root and tower base cross sections in the structural model of the DTU 10-MW reference wind turbine.

**Table 2.** Blade root and tower base section parameters of the DTU 10-MW model in HAWC2

| Cross section | radius (m) | In-plane moment of inertia (m$^4$) |
|:---:|:---:|:---:|
| Blade's root | 1.765 | 4.837 |
| Tower base | 4.15 | 8.416 |

Using Eq. (9) and replacing the stresses with the corresponding load/moment amplitudes, Eq. (10) can be rewritten as Eq. (12).

$$D = (\frac{c}{I_y})^m \sum_{i=1}^{Ns} \frac{n_i M_{x_i}^m}{k} \tag{12}$$

The parameter $M_{x_i}$ in Eq. (12) represents the load (moment) ranges in the time series. We use Eq. (12) to estimate the DEL, a parameter that we use in the current study to represent damage. The following section describes DEL.

### 2.4.1 Damage equivalent load

DEL is a tool to compare the damage caused by different variable amplitude loading scenarios. Burton et al. (2011) define DEL as the magnitude of the constant-amplitude load or stress causing the same damage as the variable-amplitude loading with the same equivalent number of cycles ($N_{eq}$). Equation (13) shows the mathematical expression of this definition based on Miner's rule (Eq. (12)).

$$\frac{N_{eq} DEL_s^m}{k}(\frac{c}{I_y})^m = \frac{\sum_{i=1}^{Ns}\left(n_i M_{x_i}^m\right)}{k}(\frac{c}{I_y})^m \tag{13}$$

In the above Eq. (13), $DEL_s$ is the DEL of a sample of 10-minute time series containing $N_s$ number of stress bins, and $N_{eq}$ is the reference number of cycles. In the current study, we set $N_{eq}$ equal to 951 cycles corresponding to the average number of cycles in a 10-minute interval based on the simulations of DTU the 10-MW turbine with DLC 1.2 condition and frequency of sampling equal to 100. Both sides of Eq. (13) represent the expectation of damage in a time span of 10 minutes. We use the expression on the right side to simplify the reliability assessment procedure and to be able to separate the variability of the load from material properties. Equations (14) and (15) are used to calculate the lifetime damage estimation ($DEL_{lifetime}$) from the 10-minute sample DEL estimations.

$$(DEL_{bin})^m = \sum_{s=1}^{SS} \frac{(DEL_s)^m}{SS} \tag{14}$$

In Eq. (14), $SS$ is the number of 10-minute samples with the wind condition (same mean wind speed and turbulence) but with different wind realizations (different turbulence seeds). Furthermore, $DEL_{bin}$ is the DEL estimation in each wind condition (wind bin). In all study cases, $SS$ is equal to 6 samples bootstrapped from a database of 200 realizations (see Table 1).

$$DEL_{lifetime}^m = \sum_{bin}(DEL_{bin})^m \mathrm{P}(bin) \tag{15}$$

In Eq. (15), $\mathrm{P}(bin)$ corresponds to the joint probability of each wind condition (wind speed and turbulence). Since we are considering the marginal probability of turbulence conditioned on the mean wind speed bin, the joint probability equals the product of the two marginal probabilities. Thus, Eq. (16) is another representation of Eq. (15).

$$DEL_{lifetime}^m = \sum_{V_{bin}=v_L}^{v_U} \sum_{T_{bin}=t_L}^{t_U} (DEL_{bin})^m \mathrm{P}(T_{bin}|V_{bin})\mathrm{P}(V_{bin}) \tag{16}$$

In Eq. (16), the variables $T_{bin}$ and $V_{bin}$ represent the mean wind speed and turbulence in each wind bin, and $\mathrm{P}(V_{bin})$ is the probability of occurrence of each mean wind speed (see Eq. (1)). In addition, the parameters $v_L$ , $v_U$, $t_L$ and $t_U$, represent the lower bound and higher bound for mean wind speed and turbulence in each wind bin, respectively. Furthermore, $\mathrm{P}(T_{bin}|V_{bin})$ is the conditional probability of each turbulence in each mean wind speed bin.

In study case 1 (constant turbulence), the probability of the representative turbulence value is assumed to be unity. The following section presents the definition, mathematical relations, and procedure for estimating fatigue reliability.

### 2.4.2 Fatigue reliability assessment

Structural reliability is the ability of a structure to fulfill the structural design request for a defined period (ISO 2394, 2015). Equation (17) shows the probabilistic representation of this ability as a function of time.

$$R(t) = 1 - P_f(t) \tag{17}$$

In Eq. (17), $P_f(t)$ is the probability of failure at time $t$ and can be stated as the probability of exceeding a certain level. Commonly, this problem is referred to with limit state function $(g(x,t))$. The safe region is where the limit state function is positive. In the design phase, the designer sets a level of reliability for the end of the design life, which accounts for an optimal balance among failure consequences, cost of operation and maintenance, material costs, and the probability of failure tolerable by societies (Sørensen, 2015). In the present work, we perform the probabilistic reliability assessment by taking the following steps as described in (Madsen et al., 2006):

1. Modeling of limit state equation $g(X)$.

2. Quantification of uncertainties and modeling by stochastic variables $X$.

3. Applying reliability methods to estimate the probability of failure (first-order reliability method in the current case)

The limit state function in assessing safety within time $t$ can be written as Eq. (18).

$$g(X,t) = R(X,t) - S(X,t) \tag{18}$$

In Eq. (18), $R$ indicates the resistance of the component, and $S$ is the loading. In addition, $X$ represents a set of stochastic variables involved in each. In the present study, we assess the reliability in time intervals of 1 year, and the time remains constant through each reliability assessment. Thus, from here on we eliminate $t$ from the relations and notations for simplification. Failure occurs if the function $g$ in Eq. (18) is smaller than or equal to zero. In other words, the probability of failure is the probability of the limit state function being equal to or less than zero. Accordingly, the probability of failure can be defined as Eq. (19).

$$P_f = \int_{g(x) \leq 0} f_x(X) \tag{19}$$

Miner's rule does not consider the load sequence effect in variable loading and thus leads to errors in fatigue damage prediction. There are some studies (Schaff and Davidson, 1997; Yanan et al., 1991; Rognin et al., 2009) showing high errors in the fatigue estimation of the composite materials, often in the form of overestimation, when using Miner's rule. We account for the uncertainty in Miner's rule by defining the damage limit as a random variable with a mean value of 1 (limit for failure). With such an assumption, the limit state function for fatigue failure can be specified as Eq. (20) (Márquez-Domínguez and Sørensen, 2012).

$$g(X) = \Delta - D \tag{20}$$

In Eq. (20), $\Delta$ represents the fatigue limit in Miner's rule as a random variable with a mean value equal to unity. Using Eq. (13) for defining the damage, Eq. (20) can be rewritten as Eq. (21).

$$g(X) = \Delta - \frac{NeqDEL_{lifetime}^m}{k}(\frac{c}{I_y})^m \tag{21}$$

The limit state function in a specific time can be shown via expressions other than the common form of Eq. (20). Equation (22) presents one such alternative (Dimitrov, 2013). In the present work, we use Eq. (22) since it facilitates the separation of the fatigue loads from the material properties. In addition, the linearized version makes the use of simple fatigue reliability estimation methods possible.

$$G(X) = log(\frac{R(X)}{S(X)}) \tag{22}$$

Combining Eqs. (21) and (22), the limit state function in the current study is expressed as Eq. (23).

$$G(X) = log(\Delta) - log(Neq) - mlog(\frac{c}{I_y}) + log(k) - mlog(DEL_{lifetime}) \tag{23}$$

The parameters $log(Neq)$ and $mlog(\frac{c}{T_y})$ in Eq. (22) are constants. Thus, Eq. (22) consists of three random parameters related to the linear damage accumulation model ($log(\Delta)$), material resistance ($log(k)$), and load ($log(DEL_{lifetime})$).

320     To find the probability of failure, after defining the limit state function, the integration in Eq. (19) must be solved. This integration is hard to solve analytically. There are established methods for estimating the integral's result. Some of the commonly used methods are the first-order or second-order reliability methods or Monte Carlo (MC) simulations (see Melchers and Beck (2018) for further details about each method).

Veers (1990) showed that the probability of failure of a wind turbine blade joint with a design life of 20 years can vary from 325   2.2% when using the second-order reliability method (SORM) to 1.8% when using FORM. In addition, Toft et al. (2011) showed the differences in the results of MC and FORM in the case of blades can be very different because of high nonlinearities in the case of higher fatigue exponents as in composites. We also observe the differences in the MC and FORM for two scenarios (see appendix), showing that in both high fatigue exponents and low probabilities of failure, FORM is less accurate. In the current work, we have simplified the formulation of the limit state function to a linear summation of the random variables 330   to decrease such errors. In addition, we are comparing different scenarios and are not interested in the absolute values. However, we investigate the highest probabilities of failure in the case of blade and tower (see appendix). We use the FORM in the current work for reliability assessment and for defining the importance of the inputs. The next section contains more details about this method.

### 335   2.4.3   FORM and importance ranks

As stated in the previous subsection, the FORM is one of the ways to estimate the solution of the integration in Eq. (19). In this method, the problem of the limit state function being more or less than zero is redefined in the standard normal space. In other words, all the distributions of the random variables are transformed to standard normal distribution, and the expression of the limit state function is also transformed. In the standard normal space, the probability of failure problem will change into look- 340   ing for an optimum design point ($X^*$ or correspondingly $U^*$ in the standard normal space) that lies on the curve of $g(U) = 0$ and has the minimum distance from the origin. The corresponding distance is known as the reliability index ($\beta$). The reliability index is commonly used as a measure of structural reliability (for more details about FORM, see Melchers and Beck (2018)).

Equation (24) shows the relationship between the reliability index and the probability of failure (Gulvanessian et al., 2012).

$$\beta = -\Phi^{-1}(P_f) \tag{24}$$

The operator $\Phi^{-1}$ shown in Eq. (24) corresponds to the inverse CDF of the standard normal distribution.

ISO 2394 (2015) presents the basic recommendation concerning a required reliability level in terms of the reliability index in a certain reference time. The minimum required reliability index is known as target reliability. Based on (IEC 61400-1, 2005), a target value for the nominal failure probability for structural design for fatigue failure mode of the wind turbine components

in a reference period of 1 year is $5 \cdot 10^{-4}$ corresponding to target reliability of 3.7 according to Eq. (24). More specifically, Veldkamp (2007) performed a cost-benefit analysis and reported the optimal reliability level for the blade to be 2.7 (probability of failure of $3.5 \cdot 10^{-3}$) based on the analysis. In the present work, we study the sensitivity of reliability to different variables and not the levels. However, we also present the reliability levels in different study cases.

To apply FORM analysis, we first fit distributions to the estimations of $log(DEL_{lifetime})$ obtained from bootstrapping and
calculated via Eq. (16) based on 10-minute simulations. It is more realistic to assume that different materials will have a different range of $\Delta$ because of differences in the scatter of the strength data, which will, in turn, result in different distribution parameters (Le and Peterson, 1999). We gather and reuse the information about the distributions and statistical parameters of the material and Miner's rule limit from the literature.

Since the DTU 10-MW turbine is not designed against fatigue, we observed low reliability levels in the blade root and the
tower base (failure occurrence in the tower base). To lower the errors in FORM in the case of the blade (high fatigue exponent and thus high nonlinearity), we calibrate the material strength towards low probabilities of failure for the sake of accuracy. Thus, in the current study, we increase the material fatigue strength proportional to the high fatigue loads while keeping the corresponding coefficient of variation (CoV) the same as for real material to avoid effects on the sensitivity analysis. These changes will affect our reliability levels. However, the main interest in the current study is the sensitivities and changes, not the
values.

Table 3 shows the distribution parameters plus the references for the coefficients of variation.

**Table 3.** Characteristics of the material and model variables

| Variable | Component | Distribution | Mean | Standard Deviation | Reference |
|---|---|---|---|---|---|
| $log(\Delta)$ | blade | Normal | -0.1116 | 0.4724 | (Toft and Sørensen, 2011; Stensgaard et al., 2016b) |
| | tower | Normal | -0.0431 | 0.2936 | (Stensgaard et al., 2016b) |
| $log(K)$ | blade | Normal | calibrated | 0.528 | (Mortensen et al., 2023; Toft and Sørensen, 2011) |
| | tower | Normal | calibrated | 0.2 | (Sørensen, 2015; Slot et al., 2019; Toft and Sørensen, 2011) |

We would like to see the sensitivity of fatigue reliability to changes in fatigue loads in addition to material strength. Different materials (used in different components) have different fatigue exponents, and thus the effect of change in their loading on
reliability can be different (the higher the fatigue exponent, the higher the effects of loads in the overall damage and reliability). We want to take this fact into account while being consistent in the idea of the variability of $k$. Thus, we consider $k$ as the variable representing material uncertainty and redo the assessments in three different levels of $m$ in each load channel under study. We calibrate the initial reliability in the annual reliability assessments in case 1 to avoid misinformation due to the correlation between $m$ and $k$. The linearized formulation of the limit state function makes this separation easier by showing
the $m$ on the load site.

After specifying all the distributions and probabilistic parameters of each random variable in Eq. (23), we transform each non-normal distribution to normal using the normal-tail approximation and Rackwitz-Fiessler algorithm (Rackwitz and

Fiessler, 1978) to find the optimum design point. The following contains the relations and procedures for the transformation and solving procedures.

In transforming non-normal continuous distributions to standard normal, since the design points $(X^*)$ are usually located at the tail of the standard normal distribution, we estimate the corresponding mean $\mu_{x_i^*}$ and standard deviation $\sigma_{x_i^*}$ based on the design point $x_i^*$ using Eq. (25) and Eq. (26), respectively (Rackwitz, 2007).

$$\sigma_{x_i^*} = \frac{\phi(u_i^*)}{f(x_i^*)} \tag{25}$$

$$\mu_{x_i^*} = x_i^* - \sigma_{x_i^*} u_i^* \tag{26}$$

In Eq. (25) and Eq. (26), the operator $\phi$ is the CDF of standard normal distribution, and $f$ corresponds to the PDF of the initial non-normal distribution. In addition, $x_i^*$ is the non-normally distributed random variable in the design point $(X^*)$, and $u_i^*$ is the corresponding element in the design point in the standard normal space $(U^*)$. $u_i^*$ is acquired as Eq. (27).

$$u_i^* = \Phi^{-1}(F(x_i^*)) \tag{27}$$

$F$ in Eq. (27) stands for the CDF of the point in the initial distribution. In fact, Eq. (27) presents the basic concept that in the
transformation process between different distributions, the probability of each point remains unvaried.

This method also provides information regarding the relative importance of each random variable or, in other words, the sensitivity of the output (reliability) to each input. A vector $\alpha$ provides such information: $\alpha$ is a unit vector defining the position of $u_*$ (the design point) and $\beta$ is its magnitude. Equation (28) shows the expression for this vector. and

$$\alpha = -\frac{\nabla g(u^*)}{|(\nabla g(u^*)|} = \frac{u^*}{\beta} \tag{28}$$

The problem is solved in an $n$ dimension space in which each dimension represents the values of one random variable. Thus, the unit vector $\alpha$ is composed of each variable's normalized magnitude in the design point. These normalized values define the share of each variable in defining the position of the design point. Therefore, the relative importance of the variables (known as the importance rank) can be shown by a factor as shown in Eq. (29).

$$Importance\,factor = \frac{\alpha_i}{|\alpha|} \tag{29}$$

In Eq. (29), as $\alpha$ is a unit vector, the denominator is equal to one, and thus $\alpha_i$ is the importance factor for variable $x_i$.

The failure probability at each point in time (year in this case) depends on the survival at the previous point (one year before). Thus, the annual reliability is useful for assessing the probability of failure at the end of each year. The annual probability of

failure in time $t$ conditional on survival in time $(t-\Delta t)$ is a special case of conditional probability. The corresponding posterior probability is shown in Eq. (30).

$$\Delta P_f(X,t) = \frac{P(X, t - \Delta t \leq f \leq t)}{P(X, f > t - \Delta t)} = \frac{P_f(X,t) - P_f(X, t - \Delta t)}{(1 - P_f(X,t))} \qquad (30)$$

Using Eq. (24), the corresponding annual reliability index is as in Eq. (31).

$$\Delta \beta(X,t) = -\Phi^{-1}(\Delta P_f(X,t)) \qquad (31)$$

### 2.4.4 Sampling

To fit the distributions to DEL data in different case studies, we need to sample from the turbulence distribution in each wind

speed bin to account for turbulence probability (see Eq. (16)). For sampling turbulence levels from cases 2 and 3, we divide their corresponding probability space into 5% probability intervals, and then we consider the median of each probability interval as the representative. We derive the corresponding turbulence sampling point by taking the inverse of the CDF at the median point. Following such an approach, the samples can account for all probability levels equally. The probability of each sampling point is equal to 5%. Fig. 3 shows the resulting sampling points in different mean wind speeds.

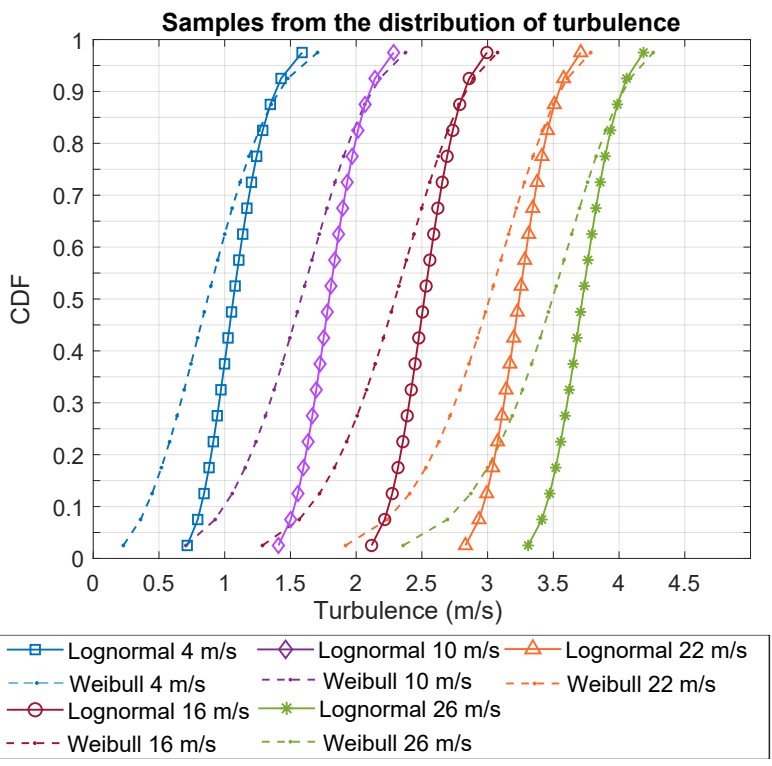

**Figure 3.** Sampling points of turbulence within intervals of size 0.05 in the corresponding in case 2 (lognormal distribution) and case 3 (Weibull distribution) at five different mean wind speeds

Figure 3 reveals that the turbulence levels within the same probability are higher in case 2 (lognormal distribution) compared to case 3 (Weibull distribution) below 90% quantile. The trend changes above 90% quantile. The differences are higher in the higher turbulence levels with lower probability, especially in higher mean wind speeds.

    The following section provides the results of the study.

## 3   Results and discussions

The current section presents the results of the study in two steps. First, we introduce the distributions of DEL in different turbulence modeling cases in Sect. 3.1. Then, in Sect. 3.2, we investigate the change of fatigue reliability through time in different turbulence modeling approaches and in different Wöhler curve exponents. Sect. 3.3 contains the sensitivity analysis of the reliability to changes in random inputs in different scenarios. Sect. 3.4 investigates the effects of the design class on the overall results of the previous sections. Finally, Sect. 3.5 includes supplementary discussions about the results. All the results

are provided for both the blade root flapwise and tower base fore-aft load channels.

### 3.1   Probability distributions and statistical parameters of load

Wind speed fluctuation is one of the main causes of fatigue damage, especially in the load channels like the blade flapwise and tower base fore-aft. A change in the wind standard deviation (turbulence) in each mean wind speed level directly changes the estimated fatigue damage. In the current section, we look into the change in the distribution of DEL with the change in

turbulence characterization approach in the IEC NTM. In case 1, each realization of $DEL_{bin}$ is based on one turbulence level (representative turbulence), while in the other two case studies, it is a result of integration over all 20 turbulence levels sampled from lognormal and Weibull distributions.

    Figure 4 shows $DEL_{bin}$ values averaged over all seeds in each turbulence level (Eq. (14)) in cases 2 and 3. Figure 4 only

contains the fatigue exponent considered in the design, equal to 10 in the blade root and 3 in the tower base.

    The bar plots of Fig. 4 show the binning over mean wind speeds and turbulence levels. It reveals the increase of the DEL with an increase in the mean wind speed and standard deviation of the wind. The only exception to this trend is the DEL in the tower base around mean wind speeds of 6 m/s and 8 m/s. The reason is that around these mean wind speeds, there is a local peak in DEL values due to resonance (Mozafari et al., 2023).

Another observation from Fig. 4 is the relatively fast decrease of the $DEL_{bin}$ in tower base as a function of turbulence compared to the blade. In other words, the difference between $DEL_{lifetime}$ obtained from a single high turbulence level and a single low level is relatively higher in the tower base. This is partly because of the resonance in the tower (see Mozafari et al. (2023)), which enhances the effect of turbulence level on fatigue loads. Thus, we expect the integration over all turbulence levels (see Eq. (16)) to be more effective in decreasing variability in $DEL_{bin}$ in the tower case. As a result, $DEL_{lifetime}$ esti-

mations in the case of the tower base would also show lower variability. The following includes observations on $DEL_{lifetime}$

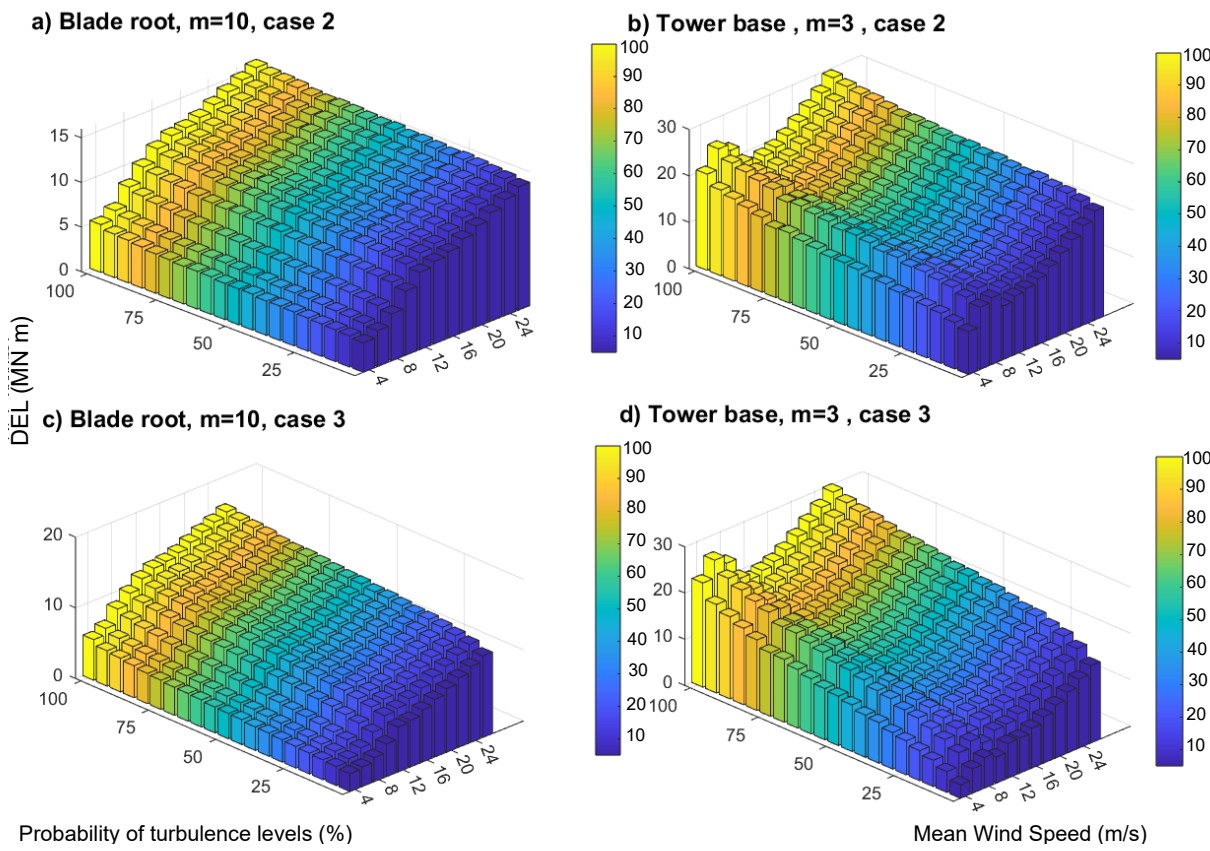

**Figure 4.** $DEL_{bin}$ in each mean wind speed and turbulence level (wind bin) in a) case 2 (turbulence sampled from lognormal distribution) for blade root, b) case 2 for the tower base, c) case 3 (turbulence sampled from Weibull distribution) for blade, and d) case 3 for tower base (the results are only shown for comparison and the DEL dimensions are relative)

distributions.

Using the 200 samples in each wind bin (consisting of constant turbulence and constant mean wind speed), we calculate the $DEL_{lifetime}$ using Eq. (15) in case 1 (constant 90% turbulence) and Eq. (16) in cases 2 and 3 (lognormal and Weibull distribution). We use a samples size of six (recommended number of samples by the IEC 61400-1) from the database and repeat 1000 times with replacement (bootstrapping) to obtain the distributions of $DEL_{lifetime}$. Figure 5 shows the probability density function (PDF) of the $DEL_{lifetime}$ estimations in different turbulence cases and different fatigue exponents in both load channels under study. The results are all normalized by the mean $DEL_{lifetime}$ in case 1.

Fig. 5 reveals that the overall variability of the $DEL_{lifetime}$ realizations are higher in the case of the tower base compared to the blade root, as expected due to the reasons discussed above and in (Mozafari et al., 2023). This remains the case for all approaches of turbulence modeling. The distributions are not significantly impacted by the fatigue exponent, as indicated by

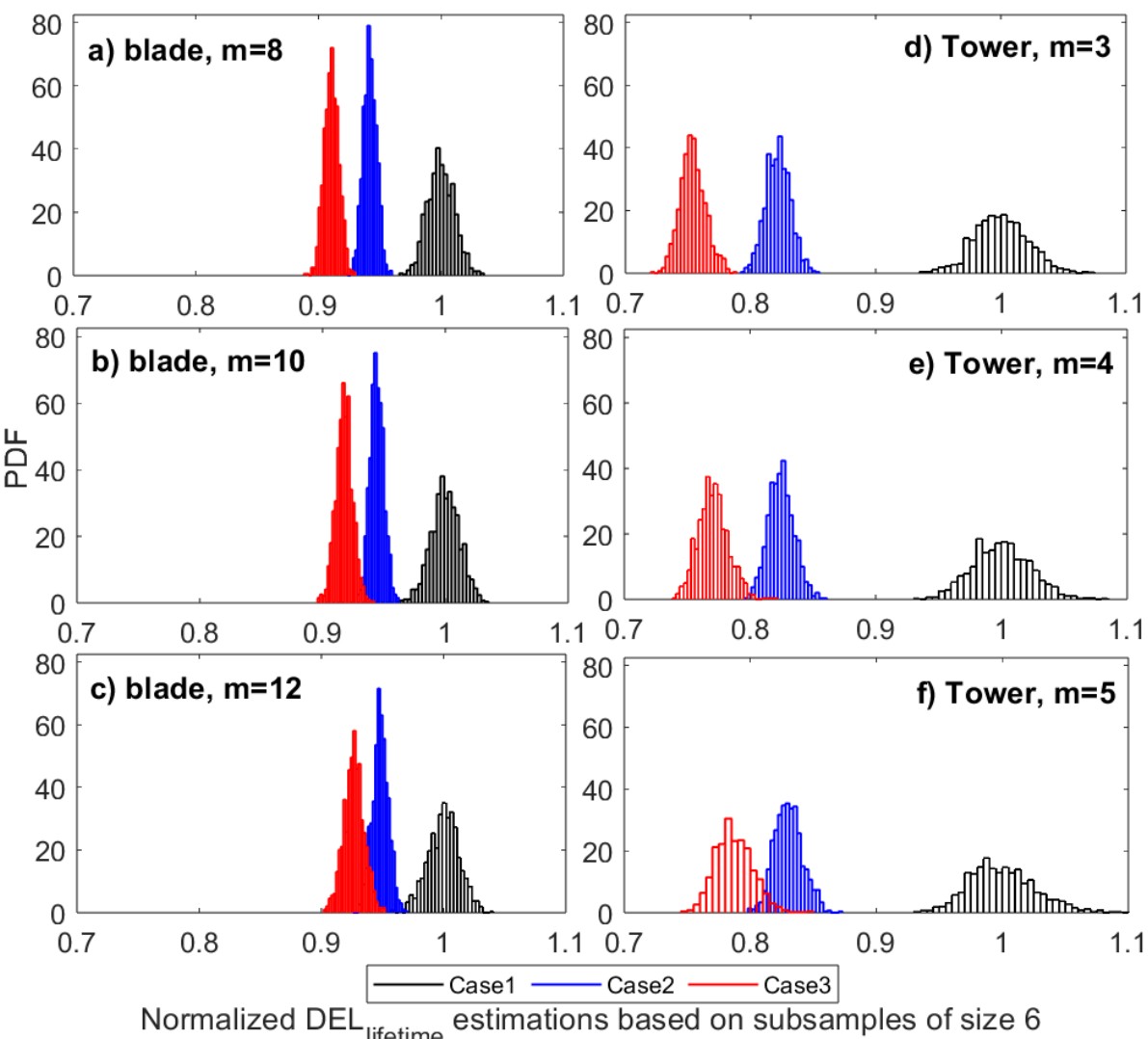

**Figure 5.** Probability density function (PDF) of normalized DEL estimates using 1000 bootstrap samples of size 6 in different turbulence study cases based on a) blade root flapwise ($m = 8$), b) blade root flapwise ($m = 10$), c) blade root flapwise ($m = 12$), e) tower base fore-aft ($m = 3$), f) tower base fore-aft ($m = 4$), g) tower base fore-aft ($m = 5$)

the similarity of the distributions in the different rows of Fig. 5.

Based on Fig. 5, the integration over turbulence distributions (cases 2 and 3) instead of using one representative value (case 1) decreases not only the mean value but also the variance of the realizations. This effect is more notable in the case of the tower base. This is expected, as Fig. 4 shows that the variability in the tower base $DEL_{bin}$ is larger than in the blade-root moment. The lower variance of the $DEL_{lifetime}$ realizations is partly because we do one extra integration step in the calculations of

this parameter in cases 2 and 3 (see (Mozafari et al., 2023) to see the effect of summation on the variability in case 1). The integration over the whole range results in an expected value, which is more robust than a single value (90% quantile in this case).


Comparing the distributions of $DEL_{lifetime}$ in case 2 (lognormal) and case 3 (Weibull) in Fig. 5 reveals that the mean levels in case 2 are lower than in case 3. In other words, estimating $DEL_{lifetime}$ based on the Weibull distribution of turbulence is less conservative than the lognormal distribution approach. This complies well with the expectations based on the characteristics of the two distributions discussed in Sect. 2.4.4. The bias in the mean value of $DEL_{lifetime}$ calculated with the

Weibull/lognormal distributions is more significant for the tower base than the blade root. Thus, using representative turbulence is relatively more conservative than the other approaches in the case of the tower base. For example, for a tower with a fatigue exponent of 3, the $DEL_{lifetime}$ estimation can vary by more than 35% when using six turbulence seeds.

Although the Weibull distribution of turbulence generally results in lower DEL estimations, there is an overlap between

distributions of all cases in the blade root. There is also an overlap between cases 2 and 3 in the tower base. This means if the designer uses only six realizations, there is a chance that the $DEL_{lifetime}$ estimation using the Weibull distribution is more conservative than the lognormal. For example, in the tower base with a fatigue exponent of 5, $DEL_{lifetime}$ estimations in case 3 can be around 5-7% more conservative than in case 2. In addition, when using six seeds in the blade root, lognormal can provide a less conservative DEL estimation than the 90% turbulence level model. Although such occurrences are rare, the

chances increase with an increase in the fatigue exponent in each load channel.

The general conservatism of using a 90% turbulence level in the DEL evaluations can show itself in the fatigue reliability estimations. In addition, since changing the method of modeling the turbulence changes the standard deviation of the DEL realizations, the sensitivity of the reliability levels to the DEL changes can also vary from one case to another. We study the

extent of such an effect in different fatigue exponents and in different load channels in the next subsections.

## 3.2 Fatigue reliability in different cases

In the reliability assessments, we use the $log(DEL_{lifetime})$ as the parameter representing the fatigue load (see Eq. (23)). We must determine an appropriate probability distribution for the load parameter to complete the reliability analysis. Thus, we first find the probability distributions of $log(DEL_{lifetime})$ in different conditions (each condition includes a load channel, a normal

turbulence modeling approach, and a specific fatigue exponent). Tables 4 and 5 represent some of the best distribution fits to the $log(DEL_{lifetime})$ data and their parameters in three different turbulence model cases in $m = 10$ and $m = 3$ for the blade root and tower base, respectively (for more cases see appendix). We find the best distribution fits among different options (GEV, lognormal, normal, and Weibull in this case) using maximum likelihood method and Akaike information criterion (Akaike, H., 1973).

Using the distributions of $log(DEL_{lifetime})$ and the distributions of other parameters ($log(\Delta)$ and $log(k)$), as we previously defined in Table 3, we estimate the annual reliability and its change through the lifetime (see Eq. (31) and Eq. (30)). It should

**Table 4.** Best distribution fits to $Log(DEL_{lifetime})$ in different turbulence modeling cases considering flapwise bending moments in the blade root ($m = 10$)

| Case number | Distribution | Par 1 | Par 2 | Par 3 |
|---|---|---|---|---|
| 1 (Representative turbulence) | GEV $(\mu, \sigma, \zeta)$ | -0.299 | 0.012 | 2.405 |
| 2 (lognormal turbulence) | GEV$(\mu, \sigma, \zeta)$ | -0.239 | 0.006 | 2.349 |
| 3 (Weibull turbulence) | Normal $(\mu, \sigma)$ | 2.323 | 0.007 | - |

**Table 5.** Best distribution fits to $Log(DEL_{lifetime})$ in different turbulence modeling cases considering fore-aft bending moments in the tower base ($m = 3$)

| Case number | Distribution | Par 1 | Par 2 | Par 3 |
|---|---|---|---|---|
| 1 (Representative turbulence) | lognormal$(\mu, \sigma)$ | 1.11 | 0.01 | - |
| 2 (lognormal turbulence) | Normal$(\mu, \sigma)$ | 2.824 | 0.012 | - |
| 3 (Weibull turbulence) | GEV$(\mu, \sigma, \zeta)$ | -0.22 | 0.01 | 2.73 |

be noted that although we derive different distributions of $log(DEL_{lifetime})$ in the case of different Wöhler exponents, the distributions in Table 3 only refer to the reference levels of this exponent in the design. For the sake of comparison of the trends, we modify the mean value of $log(k)$ in different variations of fatigue exponents ($m = 4, m = 5$ for the tower and $m = 8, m = 12$ for the blade) such that we get the same reliability level in the first year in case 1. The modification sets a benchmark for comparison. Figure 6 shows the reliability change over 20 years in the blade root and the tower base in different conditions.


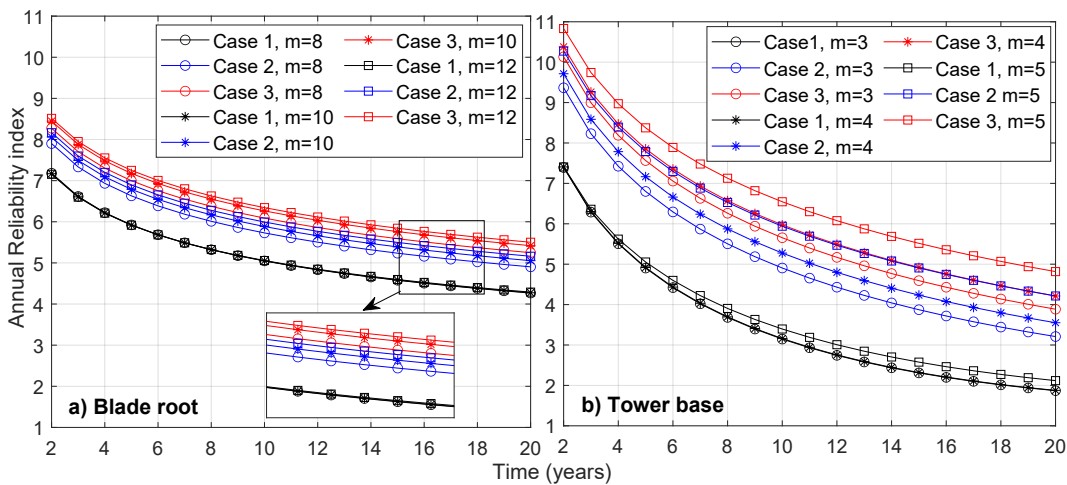

**Figure 6.** Reliability index through the lifetime considering a) blade root flapwise and b) tower base fore-aft moments in cases 1, 2, and 3 considering different Wöhler exponents

The results in Fig. 6 show that in both load channels and in different fatigue exponents, constant turbulence (black lines) provides conservative annual reliability levels. The next conservative evaluation belongs to the lognormal distribution (blue lines) and the last to the Weibull distribution (red lines). The ranks remain the same in all cases. In other words, the overlaps in the distributions of $DEL_{lifetime}$ observed in Fig. 5 between cases 2 and 3 do not affect the reliability trends. We explain the reason in the sensitivity analysis presented in the next section.

In general, the tower base shows a very fast reduction in reliability over time. One possible reason is the high mean value of the $log(DEL_{lifetime})$ versus material fatigue resistance $(log(k))$ in this load channel. We are using the conventional method of linearly scaling the damage with time. This method leads to a fast increase in damage over time in a high DEL magnitude.

In the case of constant turbulence in the tower base, the change in the fatigue exponent $(m)$ has a more visible effect on the rate of reliability declination through time. In this case, a larger $m$ increases the decline in reliability level from the same initial point at year 2.

Notice that the difference in the results in the tower base are more visible because of the higher initial variability in the DEL in this load channel. In the case of the blade, the same trends occur, but they are less visible.

## 3.3  Importance ranks of the inputs

We study the sensitivity of the reliability level at year 20 to each random input to the limit state function. The importance rank of each of the inputs is derived from FORM analysis (see Eq. (29)). Figure 7 and Fig. 8 show the relative importance levels in different cases and fatigue exponents for the blade root and tower base, respectively. In these plots, the extent of differences does not represent the absolute sensitivity to the load, material fatigue strength, and Miner's rule but their relative importance. The reason is that the random inputs to the model are in logarithm scale and showing a different CoV for each variable. The change in the percentages from one case scenario to another is still a good measure for comparing sensitivity of the reliability to each source of uncertainty and tracking changes with change of scenarios.

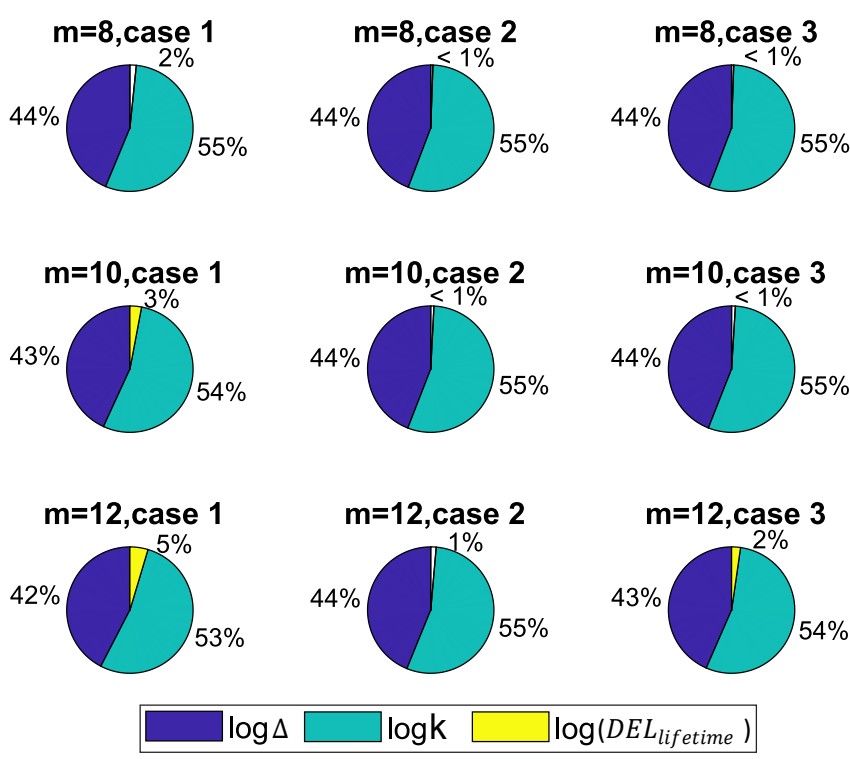

**Figure 7.** Importance factors of different random inputs in assessing annual reliability level at year 20 in blade's flapwise load channel in cases 1, 2, and 3 considering different Wöhler exponents

Figure 7 shows the relatively high importance of the fatigue resistance of the material in the case of the blade root. The second importance belongs to the uncertainty in fatigue accumulation model. The relative effects of the two are much higher than the loads because of the low variance of load in the blade. The lower variance in the DEL when using integration over turbulence in each wind speed bin (as shown in Fig. 5) decreases the effect of this parameter on the reliability level.

The share of the logarithm of load in the overall reliability increases with the increase of fatigue exponent, as we expect. However, the importance of the load uncertainty compared to the other two parameters is negligible in all cases.

Figure 8 shows the relative high importance of the Miner's rule uncertainty as the coefficient of variation in $log(k)$ in steel (material in the tower base) is much lower than in the composite (in the blade). The second effective parameter is the fatigue resistance of the material, and the load is the least important part of the uncertainty in the reliability.

The effects of the fatigue loads on the reliability in the tower base are relatively higher than in the case of the blade root (see Fig. 7). This is because of the higher CoV in the fatigue loads in the tower base.

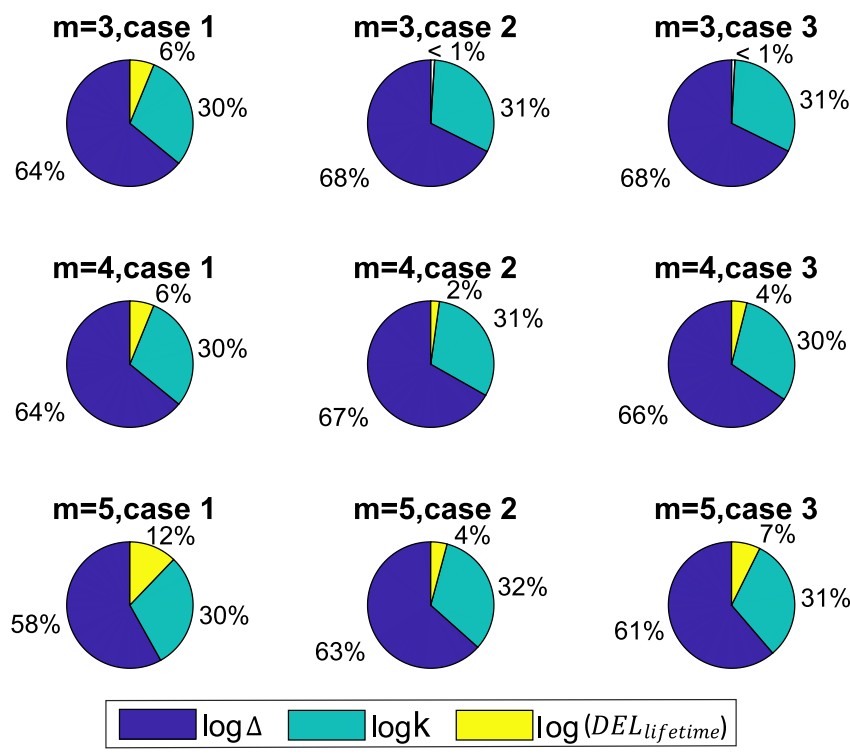

**Figure 8.** Importance factors of different random inputs in assessing annual reliability level at year 20 in tower base fore-aft load channel in cases 1, 2, and 3 considering different Wöhler exponents

As seen in the previous load channel (blade root flapwise), we observe an increase in the importance of the loads with an increase in the fatigue exponent. In addition, the case 2 and 3 turbulence modeling decrease the CoV of $log(DEL_{lifetime})$ and thus its importance in fatigue reliability.

According to both Fig. 7 and Fig. 8, there is no obvious change in the importance factors when changing the distribution from lognormal to Weibull. In other words, the change in CoV of DEL is very low when changing the distribution of turbulence. It must be emphasized again that the differences are exaggerated since the input parameters are in logarithmic scale; however, the relative importance is valid and reliable.

### 3.4 Effects of wind turbine design class

The thickness of the tail in the lognormal distribution is dependent on its standard deviation. The standard deviation of the distribution in different cases of NTM is a function of the reference turbulence level (see Eq. (3) to Eq. (8)). This means that there is a possibility that the results of the current study change with the wind turbine class. One case of lower reference

turbulence intensity equal to 0.1 is tested in the current study showing the same trends. Figure 9 shows the distributions of $DEL_{lifetime}$ in three different cases of the study for $I_{ref} = 0.1$ for two of the scenarios (tower base, $m = 3$ and blade, $m = 10$).

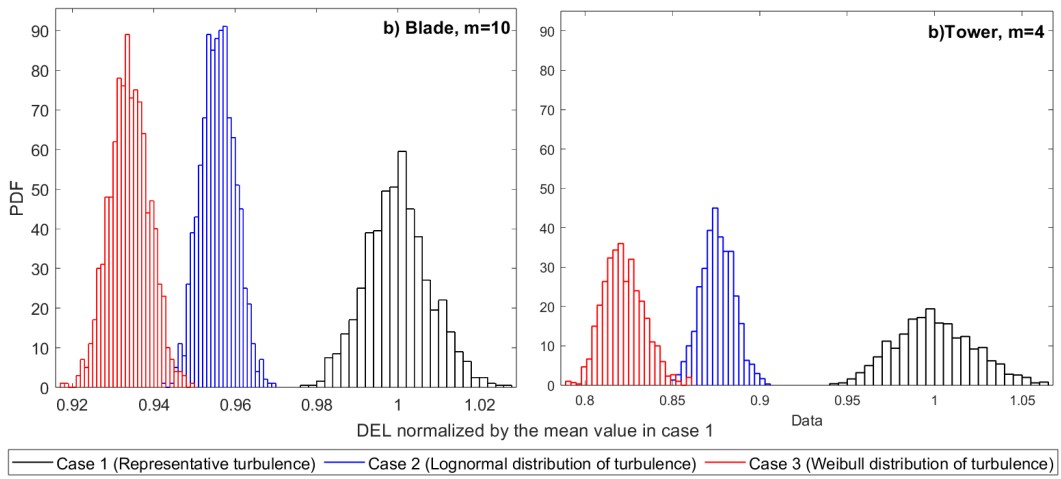

**Figure 9.** Different distributions of $DEL_{lifetime}$ in two example cases of a) blade, $m = 10$ and b) Tower, $m = 4$ for $I_{ref} = 0.1$ considering different editions of normal turbulence model

In forming the distribution of the wind speed (using the Weibull distribution shown in Eq. (1)), the annual mean wind speed defines the scale factor of the Weibull distribution. The scale factor in the Weibull distribution directly affects the standard deviation. Thus, there might be a change in the overlaps of different fatigue load data with a change in the class of the annual mean wind speeds. We study this effect via some examples with different mean wind speeds in two different reference turbulence intensities to cover some design classes. Figure 10 presents the results for one case with the highest possibility of overlap (high fatigue exponent).

Figure 10 reveals that with a decrease in the annual mean wind speed, the overlap between the DEL distributions resulted from different NTM case increases. As an example, there is a visible overlap between the long-term fatigue load distributions in the case of using representative turbulence and using full Weibull distribution. This means that in design classes with very low mean wind speed, when using six turbulence realizations, the designer can possibly get more conservative results from following edition 4 NTM versus edition 1 of the IEC standard.

According to Fig. 10, there is no obvious mutual effects from changing both representative turbulence and mean wind speed, while the changes due to the change in the mean wind speed are more obvious. To make sure that there is no such consideration in the case of the tower with the highest $m$ ($m = 5$), this case is investigated with low reference turbulence and low annual mean wind speed. Figure 11 shows the corresponding results.

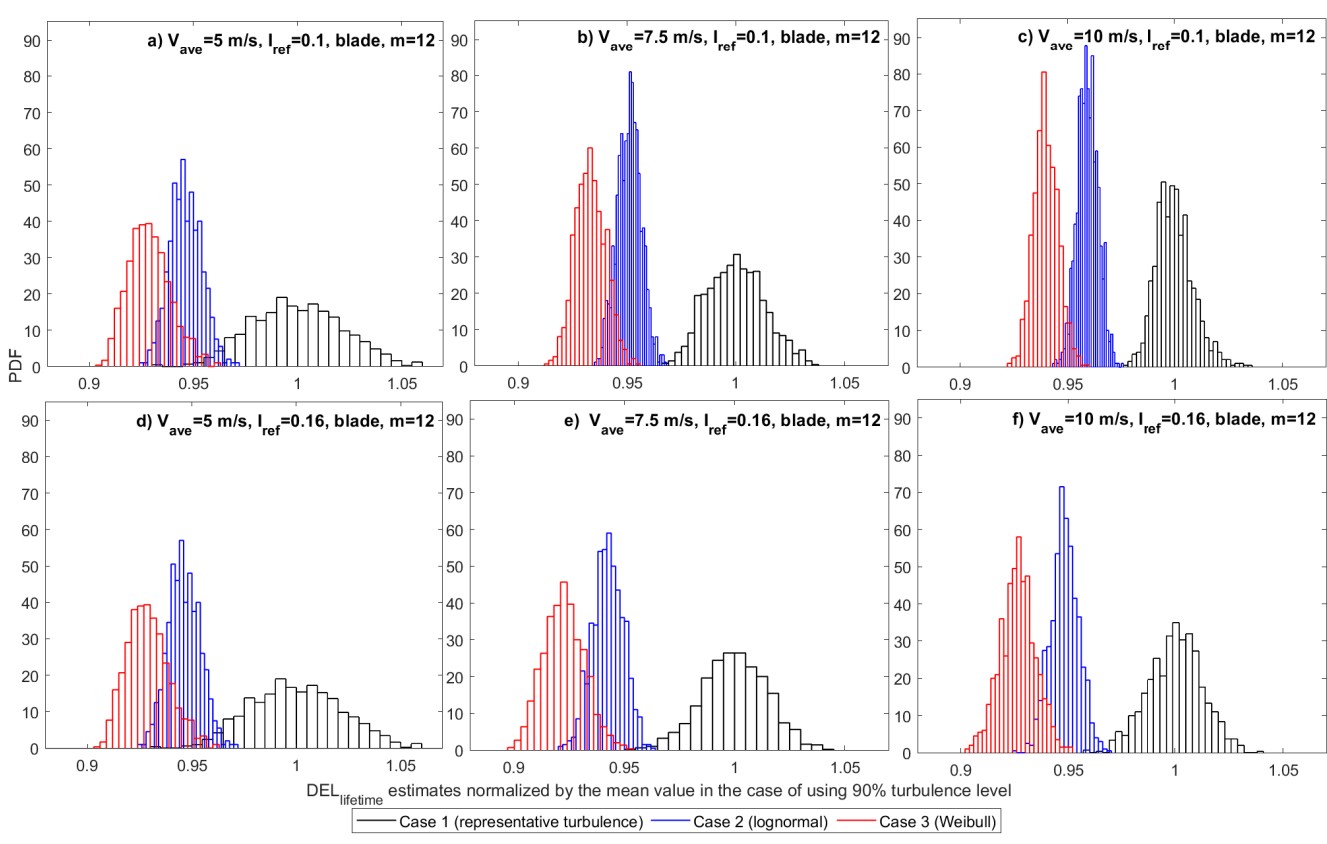

**Figure 10.** $DEL_{lifetime}$ probability distributions using different editions of NTM in different annual hub-height mean wind speed and reference turbulence intensity considering blade root flapwise moments, $m = 12$

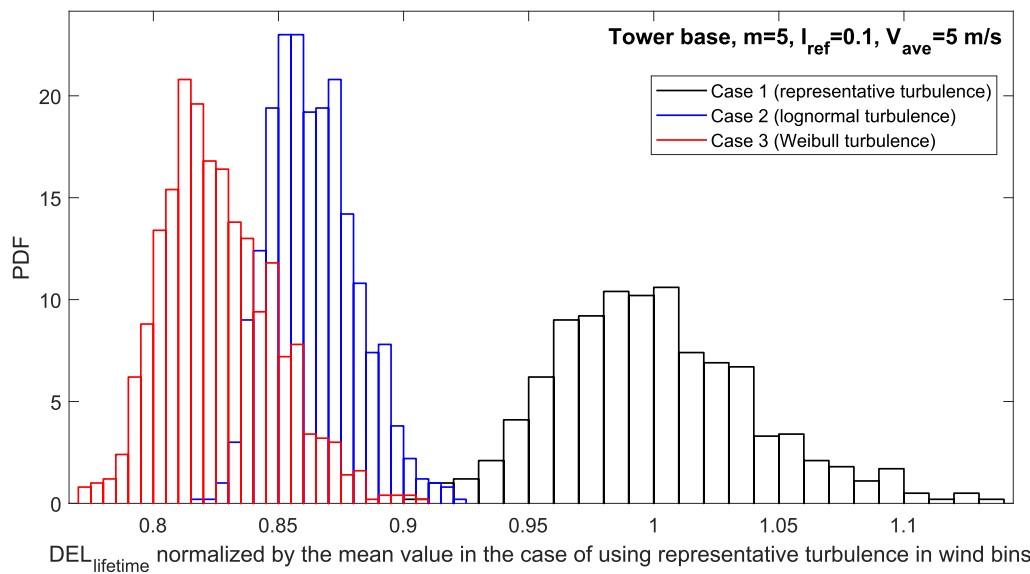

**Figure 11.** $DEL_{lifetime}$ probability distribution obtained from using different editions of normal turbulence model in annual hub-height mean wind speed of 5 m/s and reference turbulence intensity of $0.1$ considering tower base fore-aft moments, $m = 5$

As Fig. 11 shows, there is no considerable overlap in the case of the tower at the extreme end of case study design classes.

To investigate the effects of the design class on the annual reliability, two of the cases for the blade with $m = 12$ are compared in Fig. 12.

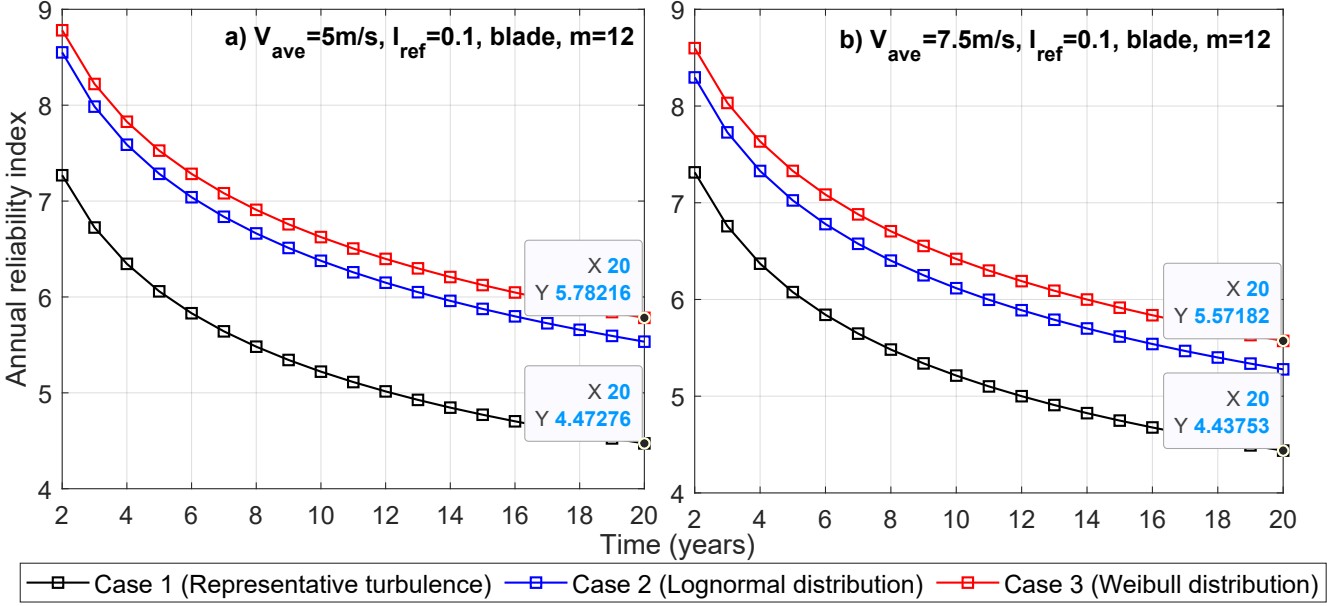

**Figure 12.** Different distributions of $DEL_{lifetime}$ in two example cases of blade, $m = 12$ in two annual average mean wind speeds of 5 m/s and 7.5 m/s considering different editions of NTM

The results presented in Fig. 12 show a higher difference between the annual reliability level at the end of the design life in an IEC design class with a lower annual mean wind speed. In such classes, although the mean levels get closer between case 1 and case 3, the higher variability in case 3 decreases the corresponding reliability level when using a probabilistic approach. It is important to consider this difference in case of any further calibration of the safety factors in future versions of the standard.

## 3.5 Overall discussions

The results (especially in Fig. 6) reveal the high effect of the fatigue exponent in reliability estimations considering tower base fore-aft moments due to the relatively high magnitude of the fatigue load in this load channel. The fatigue strength curves of the steel structures are normally bilinear. The selection of the fatigue exponent at the design level, when using damage equivalent loads, can vary between the lower, the higher, or the average slope of the actual S-N curve. The results in the current work elaborate on the effect of the selection of the steel fatigue exponent at the design level, as the annual reliability levels of the tower base (normally experiencing high loads) can be very sensitive to this parameter.

In addition, the results of reliability estimations show that the small overlaps seen in the DEL distributions are not very important when coming into the reliability framework. This has been made clearer in the sensitivity analysis, where the effects of load variations are shown to be relatively negligible.

All in all, although the choice of the turbulence characterization approach impacts the reliability and the sensitivity of the reliability to different uncertainty sources, these effects are not notable when compared to the high sensitivity of the reliability to other sources in every case. In other words, accurate modeling of the damage accumulation or more accurate characterization of the material properties (especially in the case of composites) can impact the reliability levels to a relatively higher extent. There is a 50% change in the reliability level at year 20 in the case of the tower base with $m = 5$ with change in the loads. Considering the higher effect of the changes in the material properties, the effects of uncertainty in this random variable on reliability can be drastic.

There are also some limitations and potential for extension of the study. The most important assumptions that can limit the generality of the results are as below:

1. Reference turbulence intensity (design class of the wind turbine): The thickness of the tail in the lognormal distribution depends on its standard deviation. The standard deviation of the distribution in different cases of NTM is a function of the reference turbulence level (see Eq. (3) to Eq. (8)). This means that there is a possibility that the results of the current study change with the wind turbine class. Some examples are tested in the present study. However, we encourage similar studies on other classes of wind turbines to track the possible differences in the trends and results. In addition, the annual mean wind speed influences the variability of the long-term fatigue loads, and thus overlaps between different study cases. We suggest further studies on the lower mean wind speeds and the combination of the class with other changes.

2. Additional averaging of data in case of full distributions: A potential concern with the results is the difference in sample size for the different cases. There are more 10-minute simulations involved in estimations in cases 2 and 3. A larger sample size naturally decreases the variance in the $DEL_{lifetime}$ evaluations due to the law of large numbers (see (Mozafari et al., 2023) for more details). To investigate whether this is significant, we checked the effect by using different combinations of seed numbers, and the corresponding effect on the trends is negligible. However, we encourage testing of

different calculation approaches to track any possible changes in the variability of fatigue loads in cases of using full distributions of the turbulence.


3. Other load cases: Among the standard design load cases related to fatigue, idling, and power production with fault also include the NTM in the IEC standard. It is valuable to perform the same study considering these other load cases and their corresponding probabilities. In addition, considering all relevant load cases for fatigue (including shutdowns and start-ups) can change the long-term fatigue distributions and trends and should be considered in future studies.


4. Specificity of the wind turbine response: The main study uses the DTU 10-MW wind turbine as the case study. The size and design of the wind turbine and its controller's design affect the turbine's response to a specific wind input. The Siemens 2.3-MW wind turbine (with a smaller size) but a similar controller and class is checked, and the results show the same trends in distributions of the long-term fatigue load. In future studies, testing other wind turbines with a different type of controller and also using other aeroelastic simulation tools is beneficial.


5. Variability of the material properties and damage accumulation rule: The variability of the initial material fatigue strength and Miner's rule are taken from the literature. Updating the corresponding coefficients of variations can change the levels in the sensitivity analysis.


6. Variability of the fatigue loads: The only variable considered in defining the fatigue loads is the variation in the turbulence inputs. While the shares of load uncertainty due to this specific variation are covered, the sensitivity results can vary when considering other sources of uncertainty in the loads.


7. Method of reliability assessment: The first-order reliability method performs well in very low probabilities of failure and less accurately in higher failure probabilities. Doing the same reliability analysis using MC instead of FORM can provide more accurate reliability estimates if computational resources are available (see appendix for a detailed explanation regarding computational expenses of MC).


8. Offshore versus onshore: The study uses aeroelastic simulations with only onshore wind inputs. However, in the offshore cases, the effects of wind turbulence on the structure response varies. The effect is more in the case of tower loads. We recommend performing the same study for offshore cases to investigate the possible changes in the trends.

# 4  Conclusions

The assessment of the remaining fatigue lifetime hinges upon reanalyzing the reliability. In performing such an analysis per the IEC standards, a designer can choose to follow different recommendations regarding probabilistic modeling of the turbulence. The ramifications of those choices are currently unclear. For example, as the present study shows, using six realizations for the estimation of DEL, in the case of the tower with a fatigue exponent of 3, the estimations based on the Weibull distribution of turbulence can differ by 40% from the representative turbulence approach. Regarding fatigue reliability at the end of the design

life (20 years), the differences are up to 50% in the case of the blade root and up to 200% in the case of the tower base. Such high difference can change the possible scenarios at the lifetime extension stage and must be considered. The difference in reliability levels varies with a change in the design class. Consideration of the difference between classes is important in case of any further calibration of the safety factors in future versions of the standard.

The results presented in this paper are applicable to the wind turbine design stage. The study informs the designer about the extent to which following different editions of the IEC standard can change the expected value and uncertainty of the fatigue damage evaluations based on turbulence input. It also shows how the annual fatigue reliability and sensitivity of reliability change in load channels of interest and different fatigue exponents.

Furthermore, the reliability estimation is based on a simplified linearized limit state function, making the complete separation of the loads from material properties possible. If all the random variables are lognormally distributed (highly possible), this formulation results in a very simple and fast reliability analysis at the design level.

The importance ranks of the variables reveal that although the change in the turbulence characterization changes the dis-
tribution of the fatigue loads and the fatigue reliability, focusing on decreasing the material or models' uncertainty is more effective. This is due to the relatively high uncertainty in the material properties and linear damage accumulation rule.

Using Monte Carlo simulations, considering other sources of uncertainty in the load, testing for other wind turbine designs and classes, using other aeroelastic tools, considering offshore cases, and finally, using different approaches for the design of
experiments are some suggestions for future studies.

*Code and data availability.*  All used data for the DTU 10-MW case study, in addition to the codes, can be found in:

https://gitlab.windenergy.dtu.dk/shmoz/phd-research-papers/-/tree/main/Paper3

## Appendix

The parameters of the best-fitted distributions to $log(DEL)$ in different load channels under study and different fatigue exponents are shown in Tables A1-D1.

**Table A1.** Best distribution fits $Log(DEL_{lifetime})$ in different turbulence modeling cases considering flapwise bending moments in the blade root ($m = 8$)

| Case number | Distribution | Par 1 | Par 2 | Par 3 |
|---|---|---|---|---|
| 1 (representative turbulence) | lognormal $(\mu, \sigma)$ | 0.822 | 0.005 | - |
| 2 (lognormal turbulence) | lognormal $(\mu, \sigma)$ | 0.795 | 0.003 | - |
| 3 (Weibull turbulence) | lognormal $(\mu, \sigma)$ | 0.780 | 0.003 | - |

**Table B1.** Best distribution fits to $Log(DEL_{lifetime})$ in different turbulence modeling cases considering flapwise bending moments in the blade root ($m = 12$)

| Case number | Distribution | Par 1 | Par 2 | Par 3 |
|---|---|---|---|---|
| 1 (representative turbulence) | Normal $(\mu, \sigma)$ | 2.510 | 0.013 | - |
| 2 (lognormal turbulence) | Normal $(\mu, \sigma)$ | 2.457 | 0.007 | - |
| 3 (Weibull turbulence) | Normal $(\mu, \sigma)$ | 2.434 | 0.009 | - |

**Table C1.** Best distribution fits to $Log(DEL_{lifetime})$ in different turbulence modeling cases considering fore-aft bending moments in the tower base ($m = 4$)

| Case number | Distribution | Par 1 | Par 2 | Par 3 |
|---|---|---|---|---|
| 1 (representative turbulence) | lognormal $(\mu, \sigma)$ | 1.176 | 0.007 | - |
| 2 (lognormal turbulence) | GEV $(\mu, \sigma, \zeta)$ | -0.218 | 0.012 | 3.044 |
| 3 (Weibull turbulence) | GEV $(\mu, \sigma, \zeta)$ | -0.180 | 0.015 | 2.975 |

Figure B1 shows the distributions of $DEL_{lifetime}$ in two different cases of the study (Ed. 1 and Ed. 4) for the Siemens 2.3-MW.

Table C1 shows the comparison of the probability of failure using FORM and MC in two scenarios (blade root $m = 10$ and tower base $m = 3$).

**Table D1.** Best distribution fits to $Log(DEL_{lifetime})$ in different turbulence modeling cases considering fore-aft bending moments in the tower base ($m = 5$)

| Case number | Distribution | Par 1 | Par 2 | Par 3 |
|---|---|---|---|---|
| 1 (representative turbulence) | GEV $(\mu,\sigma,\zeta)$ | -0.169 | 0.023 | 3.396 |
| 2 (lognormal turbulence) | GEV $(\mu,\sigma,\zeta)$ | -0.201 | 0.013 | 3.216 |
| 3 (Weibull turbulence) | GEV $(\mu,\sigma,\zeta)$ | -0.194 | 0.017 | 3.162 |

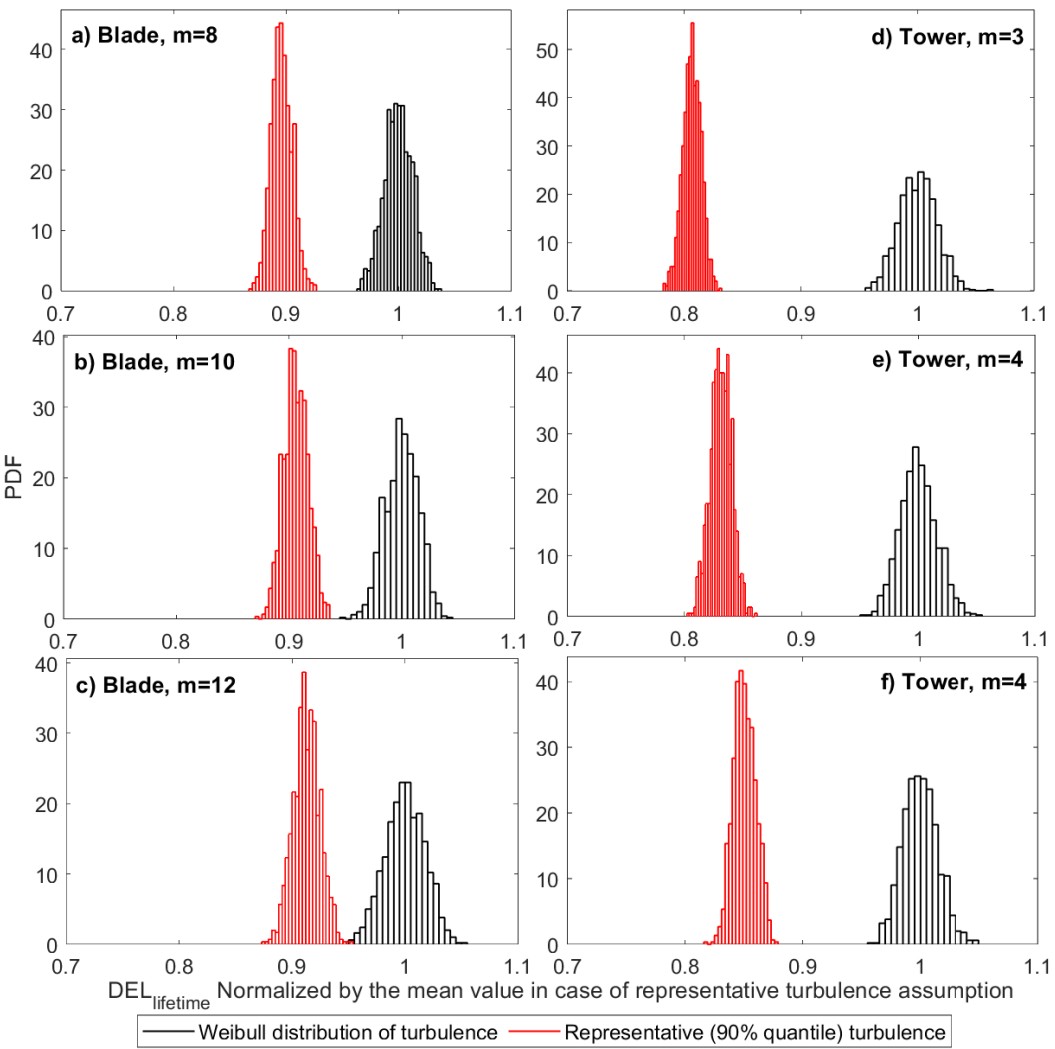

**Figure B1.** Different distributions of $DEL_{lifetime}$ considering 90% quantile of lognormal distribution and full distribution of Weibull for modeling the turbulence using the Siemens 2.3-MW as the case study

**Table C1.** Comparison of the probability of failure in years 10, 15, and 20 using Monte Carlo simulations and FORM for the blade ($m = 10$) and the tower ($m = 3$)

| Component (fatigue exponent) | Method | $P_f$ at year 10 | $P_f$ at year 15 | $P_f$ at year 20 |
|---|---|---|---|---|
| Tower ($m = 3$) | FORM | $9.6 \times 10^{-4}$ | $1.1 \times 10^{-2}$ | $3.3 \times 10^{-2}$ |
| | MC | $8.9 \times 10^{-4}$ | $2.3 \times 10^{-2}$ | $1.1 \times 10^{-1}$ |
| Blade ($m = 10$) | FORM | $4.2 \times 10^{-7}$ | $6.4 \times 10^{-6}$ | $3.7 \times 10^{-5}$ |
| | MC | $3.9 \times 10^{-7}$ | $5.9 \times 10^{-6}$ | $3.4 \times 10^{-5}$ |

## Computational expenses of Monte Carlo in reliability assessment of wind turbine structural components

For low probabilities of failure, such as in structural components of the wind turbines, a lot of simulations are needed for MC. In a Monte Carlo analysis with $N$ number of simulations, the coefficient of variation of the estimate ($P_f = 10^8$ in our case) is proportional to $1/\sqrt{N}$ (based on the law of large numbers). This means that if the actual probability of failure in a structural component is in the order of $10^{-x}$, approximately $10^{(x+2)}$ simulations are needed to achieve an estimate with a coefficient of variance in the order of $10\%$. Standard computers can save data with size up to $N = 10^9$, meaning we can capture the maximum probability of failure $10^{-7}$. It is possible to cluster the simulations, for example, to get 10 clusters of $10^9$ to capture $P_f = 10^8$. However, more and more loops will take a lot of time for standard processors.

*Author contributions.* SM, PV, and JR were responsible for the overall conceptualization of the study. SM wrote all the computer codes and performed all the data analysis. SM, PV, and KD were involved in the writing and editing of the manuscript

*Competing interests.* At least one of the (co-)authors is a member of the editorial board of Wind Energy Science.

Code and data availability

*Acknowledgements.* This work was authored in part by the National Renewable Energy Laboratory, operated by Alliance for Sustainable Energy, LLC, for the U.S. Department of Energy (DOE) under Contract No. DE-AC36-08GO28308. Funding provided by U.S. Department of Energy Office of Energy Efficiency and Renewable Energy Wind Energy Technologies Office. The views expressed in the article do not necessarily represent the views of the DOE or the U.S. Government. The U.S. Government retains and the publisher, by accepting the article for publication, acknowledges that the U.S. Government retains a nonexclusive, paid-up, irrevocable, worldwide license to publish or reproduce the published form of this work, or allow others to do so, for U.S. Government purposes.

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
