# Peer review of "Sensitivity of fatigue reliability in wind turbines: effects of design turbulence and the Wöhler exponent"

_Wind Energy Science, 2023_

## Referee Comment (RC1)

**"Sensitivity analysis of wind turbine fatigue reliability: effects of design turbulence and the Wöhler exponent" (Manuscript number: wes-2023-47)**

In this work, the influence of different approaches to determine turbulence values (90th percentile, log-normal and Weibull distribution) on the fatigue reliability of wind turbines is analyzed. This influence in contextualized by determining the influence of model uncertainty (i.e., uncertainty due to the assumption of a linear damage accumulation) and material uncertainty (i.e., uncertainty in the Basquin coefficient) as well.

As turbulence has a major impact on wind turbine fatigue loads and its modelling is discussed controversially, the topic of this paper is relevant to the readers of the WES journal. Moreover, most of the paper is nicely written and the argumentation is mainly clear. Nonetheless, there is a huge number of smaller points that have to be clarified or corrected. Hence, without a major revision, it is not suitable for a publication in the WES journal.

Some important points:

1) Section 2.2: Please, state the simulation time and the time you cut off at the beginning of each simulation to remove the initial transients. Otherwise, the description of the aero-elastic simulations is not complete.
2) Section 2.3.1: Please, highlight that you talk about the turbulence (i.e., the standard deviation of the wind speed) and not the turbulence intensity, which is also frequently used.
3) Section 2.4.1: Be careful, when using expected values of DEL, as $\left[E(DEL_{lifetime}^m)\right]^{\frac{1}{m}} \neq DEL_{lifetime}$. Furthermore, you use $DEL_{lifetime}$ in the following, but only introduce $E(DEL_{lifetime}^m)$ in Section 2.4.1. What exactly do you mean by $DEL_{lifetime}$? Please, be consistent and precise. Either use $E\left(DEL_{lifetime}^m\right)$ or $DEL_{lifetime}$ in the entire paper or define both.
4) L. 315: If you state that something has been investigated, you must show it. Hence, put the MC validation in the appendix.
5) Most equations are nicely derived. However, your final equations, i.e., equation 26 to 29, are not explained sufficiently. Please, clarify where they come from and what all terms mean. For example, it seems as if equation 28 has the unit "1/year" on the right side, but a probability, i.e., is unit-free, on the left side.
6) L. 419: Again, it is not clear what you are talking about. Is it $E\left(DEL_{lifetime}^m\right)$, $\left[E(DEL_{lifetime}^m)\right]^{\frac{1}{m}}$ or $DEL_{lifetime}$. Be very precise in your notation regarding DELs. Otherwise, thing become confusing.
7) Section 3.1 and all following sections: I am not sure whether it is necessary to show different values for $m$. In Section 2.4 you state that $m$ and $k$ are highly correlated and that you only consider $k$ as a random variable. I suggest that you should stick to this idea in the entire paper. This would also reduce the amount of information in the paper, which would increase the readability. However, if you have a good reason for showing different values for $m$, this is also fine. In this case, please, explain your reasoning.
8) L. 534: Again, if you state that you have conducted an analysis, you must show the results, at least in the appendix (cf. comment 4).

Several other less relevant points:

9) Please, rethink the title of the paper. It suggests that a sensitivity analysis is the focus of the paper. However, this is not the case.
10) Perhaps, you can rewrite the abstract. For me, it only became clear after having read the paper. It might help to shorten it and to highlight the main topic of the paper.
11) Fig. 1: "Turbulence standard deviation of the wind and…" is not self-explanatory. What is meant by "and…"?
12) L. 127: You state that you use the "DTU 10MW offshore turbine". However, you use its onshore version. That should be clarified.
13) It would be nice if you explain why you use 200 random seeds (and not more or less).

14) Eq. 1 and in the following: Please, be consistent in your notation, e.g., $V_{hub}$ and $v_{hub}$

15) Please, make clear that Eq. 2 and 3 are derived from equations given in IEC 61400-1, 2005.

16) Eq. 2 to 6: Please, keep the units consistent.

17) Eq. 2: I think there is a mistake in the equation.

18) Eq. 3: I think that it is supposed to be $I_{ref}$ and not $TI_{ref}$

19) Eq. 3: Closing parenthesis is missing.

20) Perhaps, it would be good to state that $T \sim LN(\mu_T, \sigma_T)$ for Eq. (2) and (3) and $T \sim Wbl(k, C)$ for Eq. (5) and (6) and to give $F(T)$ explicitly. Otherwise, the meaning of $k$ and $C$ are not fully clear.

21) Eq. 4: Please, make clear that this equation does not give the 90th percentile for each wind speed, but is only a linear regression approximating it.

22) Eq. 5: Is this equation correct? I think $I_{ref}$ has to be removed.

23) Fig. 2: The horizontal axis is the turbulence ($T$), i.e., the standard deviation of the wind speed, in m/s?

24) Fig. 2: The vertical axis is $\ln(1 - F(T))$, where $T \sim LN(\mu_T, \sigma_T)$ or $T \sim Wbl(k, C)$?

25) Eq. 7: I think that it should be $S^{-m}$. Otherwise, the number of allowed cycles would increase for an increasing load if $m$ is positive (what it is according to your definition in the following).

26) L. 227: $M_{x_i}$ is only the flapwise bending moment if there is no blade pitching. I think that it is an acceptable simplification for this work, but it should be mentioned.

27) Eq. 11: Use $I_y$ and not $I$ to be consistent, and $(c/I_y)^m$ should not stand in the denominator on the right-hand side of the equation.

28) L. 255: You state that $SS = 200$. However, you only use $SS = 6$ in the entire paper. Hence, this should be changed here. Furthermore, it might help to refer to Section 3.1 at this point to clarify why 200 random seeds are used, but only $SS = 6$.

29) Eq. 17: I think that the operator in this equation is not self-explanatory to all readers.

30) Eq. 19 and 21 (and in the following paragraph): I think it should be $E(DEL_{lifetime}^m)$

31) Eq. 19 and 21 (and in the following paragraph): Use $I_y$ and not $I$ to be consistent.

32) L. 339: What is meant by $R = 10$?

33) L. 340: Without knowing Section 3.1, the reader will wonder why $\log(E(DEL_{lifetime}^m))$ or $\log(DEL_{lifetime})$ is a distribution. Therefore, you should refer to the bootstrapping in Section 3.1 (cf. comment 28)

34) Fig. 3: Units for the horizontal axis are missing.

35) L. 386: The statement "Fig. reveals that […]" is only correct for $F(t) < 0.9$

36) L. 406 and Fig. 4: $DEL_{bin}$ or $E(DEL_{bin}^m)$? If it is $DEL_{bin}$, the "different seeds" do not make any sense. If it is $E(DEL_{bin}^m)$ the units in Fig. 4 are incorrect.

37) L. 413: The statement "This observation reveals […]" is not correct. Figure 4 does not provide any information about the scatter in each bin, as it shows average values for each bin, i.e., $E(DEL_{bin}^m)$. You probably mean that the variability is higher for varying turbulence values.

38) L. 433: I agree with the statement "The lower variance […]". However, you should demonstrate that this is actually the case by running case 1 with 6*20=120 seeds instead of 6 seeds and show these results in the appendix. Otherwise, the comparison is not fair.

39) L. 434: I do not agree with your second reason "The other reason is […]". If you consider different turbulence levels, you have low and high values. Hence, the variability should be higher compared to the case where you only use high turbulence levels.

40) Fig. 5: Horizontal axis is "normalized DEL".

41) Fig. 5: Where do I see the "best distribution fits" that are mentioned in the caption of the figure?

42) Section 3.2: How are the best fitting distributions determined? Out of which distributions is the best fitting distribution chosen? How is the goodness of the fit judged?

43) Table 4: I think it would help if an equation for the GEV is given somewhere. Otherwise, it is not clear what "Par 1", "Par 2" and "Par 3" are. Even for the lognormal and Weibull it would help (cf. comment 20)

44) L. 468: The statement "[…] that we get the same reliability level in the first year" only refers to case 1 if I am not mistaken.

45) L. 510: Which DEL is meant here?

46) L. 523 and 554: Lifetime DEL? Or which one?

47) L. 524: "This has been made more clear in the sensitivity analysis": Where exactly has it been made clearer? I do not find this.

48) L. 534: $I_{ref} = 0.1$? Or what do you mean here?

49) L. 547: You state that MC can only be done when having the computational resources. However, you only need to evaluate Eq. (21) which should not be computationally very demanding, when knowing the distributions. Running the aero-elastic simulations probably takes much more time. Hence, the use of FORM instead of MC should be discussed in more detail. Or am I mistaken and the procedure is computationally demand. In this case, please explain why this is the case.

50) L. 565: I would not call it "sensitivity analysis" but perhaps "importance ranking". You already stated in Section 3.3 that it is not really a representative sensitivity analysis.

Typos etc.:

51) As you can see in the following, there are quite a lot of typos and inconsistencies. As I have definitely not found all of them, I recommend a thorough proof reading.

52) Please, revise your citation style. It seems to be inconsistent.

53) L. 134: Remove the second parenthesis before "Larsen and Hansen, 2007".

54) Footnote 1: "in the time domain – developed in" not "in the time domain- developed in"

55) Eq. 1: $\exp(x)$ should be $e^x$ and pi should be $\pi$

56) Fig. 2: "Lognormal" and not "lognormal"

57) L. 199: $10^3$ and not 1e3.

58) L. 205 (and several times more): $k$ and not $K$

59) Eq. 9: Please, keep indices consistent, e.g., $I_y$ and not $Iy$

60) Table 2: "radius" not "radious"

61) L. 270: "Equation (15) shows" not "Equation (15 )shows".

62) L. 294: Missing citation "marquez2012fatigue".

63) L. 307: "resistance" not "Resistance"

64) L. 335 and 337: $10^{-4}$ and not e-4

65) Fig. 4: "Probability" not "Pobability" and 0-100% and not 0-20 (both horizontal axis).

66) Fig. 4: "MNm" not "Mnm" (vertical axis).

67) Fig. 4: 0 to 100% and not 0-20 (colour axis)

68) Table 4: "Par 2" and not "par2" and "Par 3" and not "par3"

69) L. 464: "Eq. (29) and" and not "Eq. (29 )and"

70) L. 466: I think that there is something missing in the statement "the distributions in 3 are […]"

71) Fig. 6: "b) tower" and not "b)tower"

72) L. 470: "show that in both" and not "show that the in both"

73) Fig. 7: Please, update the legend, e.g., $\ln(\Delta)$ and not "log Delta" and which DEL is meant here?

74) L. 569: "the" and not "The"

---

## Author Comment (AC1)

**Sensitivity analysis of wind turbine fatigue reliability: effects of design turbulence and the Wöhler exponent (Manuscript number: wes-2023-47)**

Shadan Mozafari[1], Paul Veers[2], Jennifer Marie Rinker[1] , and Katherine Dykes[1]

1 Department of Wind Energy, Technical University of Denmark, Roskilde, Denmark
2 National Renewable Energy Laboratory (NREL), Golden, CO, USA.

**Response letter to referee #1 comments (Manuscript number: wes-2023-47):**

We would like to thank the reviewer for the delicate and useful comments which helped us improve our work. All comments are addressed, and a revised version of the article is prepared. Please find the response to comments categorized as 'important points', 'less relevant points' and 'typos etc.' by the reviewer in the following. The comments of the reviewer are in black, and the blue texts are the responses from the authors.

**Some important points:**

1) Section 2.2: Please, state the simulation time and the time you cut off at the beginning of each simulation to remove the initial transients. Otherwise, the description of the aero-elastic simulations is not complete.

We added the below text to clarify:
'Simulations are performed for 700 seconds, from which the first 100 seconds are recognized as transient time and are omitted from the results. The transient time is defined by checking the time of stabilization for tower base side-side moments in high mean wind speeds (20-26 m/s), as this load channel is the one with the longest time of stabilization.'

In addition, we added two rows to table 1 defining the length of simulations and transient time.

2) Section 2.3.1: Please, highlight that you talk about the turbulence (i.e., the standard deviation of the wind speed) and not the turbulence intensity, which is also frequently used.

We now emphasize on this in 2 places in 2.3.1:

'The wind as a random process is mostly described by its mean value and standard deviation (turbulence) at each point of time and space.'

+

'The statistical parameters of the wind are correlated. In other words, the turbulence standard deviation of the wind (m/s) changes with a change in the mean level. '

3) Section 2.4.1: Be careful, when using expected values of DEL, as $[E(DEL_{lifetimem})]_{1m} \neq DEL_{lifetime}$. Furthermore, you use $DEL_{lifetime}$ in the following, but only introduce $E(DEL_{lifetimem})$ in Section 2.4.1. What exactly do you mean by $DEL_{lifetime}$? Please, be consistent and precise. Either use $E(DEL_{lifetimem})$ or $DEL_{lifetime}$ in the entire paper or define both.

Agreed! We changed all to $DEL_{lifetime}$.
We agree that in all cases $DEL_{lifetime}$ would be a realization that can be close or far from the expected value (true realization) based on sample size of $DEL_{bin}$.

4) L. 315: If you state that something has been investigated, you must show it. Hence, put the MC validation in the appendix.

We added a clear explanation of the possible differences in high and low fatigue exponents and added a table of comparisons between MC and FORM (in the appendix) for the blade and the tower.

In addition, we changed the calibration of the blade to have more accurate reliability levels in the blade. Thus, figure 5a is changed accordingly. Below is the new figure for annual reliability of the blade. The trends and the sensitivity levels did not change (as expected). Thus, fig5a is the only changed item without any further change in any other result of the paper or conclusions.

We also added below explanation to section 2.4.3:

'To lower the errors in FORM in the case of the blade (high fatigue exponent and thus high non-linearity), we calibrate the material strength towards low probabilities of failure for the sake of accuracy.'

5) Most equations are nicely derived. However, your final equations, i.e., equation 26 to 29, are not explained sufficiently. Please, clarify where they come from and what all terms mean. For example, it seems as if equation 28 has the unit "1/year" on the right side, but a probability, i.e., is unit-free, on the left side.

We added further explanations to equations 26 to 29. The added explanations clarify the meaning of importance ranks and how it shows the sensitivity of the reliability to each variable input. This is relevant to comment #9 regarding the focus of the study.

Regarding Eq. 28: The term $\Delta t$ in the dominator was extra (the mentioned reference is now omitted from the list of references to prevent misleading). A more complete formulation plus explanation of the logic behind it is provided now.

6) L. 419: Again, it is not clear what you are talking about. Is it $E(DEL_{lifetime m})$, $[E(DEL_{lifetime m})]^m$ or $DEL_{lifetime}$. Be very precise in your notation regarding DELs. Otherwise, things become confusing.

Resolved via changes based on comment #3.

7) Section 3.1 and all following sections: I am not sure whether it is necessary to show different values for $m$. In Section 2.4 you state that $m$ and $k$ are highly correlated and that you only consider $k$ as a random variable. I suggest that you should stick to this idea in the entire paper. This would also

reduce the amount of information in the paper, which would increase the readability. However, if you have a good reason for showing different values for $m$, this is also fine. In this case, please, explain your reasoning.

We would like to see the sensitivity of fatigue reliability to load changes in addition to changes in the material strength. Different materials (used in different components) have different fatigue exponents and thus the effect of change in their loadings can be different (the highest the fatigue exponent, the highest the effects of loads in the overall damage and reliability). We want to take this fact into account while being consistent in the idea of variability of 'k'. Thus, we consider 'k' as the variable representing material uncertainty and redo the assessments in 3 different levels of 'm' in each load channel under study. We calibrate the initial reliability in the annual reliability assessments in case 1 to avoid misinformation due to the correlation between 'm' and 'k'. The linearized formulation of limit state function makes this separation easier by showing the 'm' on the load site.

8) L. 534: Again, if you state that you have conducted an analysis, you must show the results, at least in the appendix (cf. comment 4).

Added additional results to the end of results section to showcase not only effects of the turbulence intensity, but annual mean wind speed and turbine type via examples.

**Several other less relevant points:**

9) Please, rethink the title of the paper. It suggests that a sensitivity analysis is the focus of the paper. However, this is not the case.

In fact, the main message of the article is the need to focus on the assessment of material properties prior to loads based on showing the sensitivity of reliability to changes in each of the two while changing the design turbulence levels. We agree that the process of sensitivity analysis is not the main focus of the study however, the main deliverable of the paper is provided via showing the relative sensitivity of reliability to each of the. All the results other than the sensitivity results are steps to get there.

We are also aware that the changes in the load side are not general and only cover the turbulence from all the variables that define fatigue loads' variability. However, this parameter is one of the main variables with very high impact on the fatigue loads but not the only one.

We tried to clarify the focus of study better in the new version of the abstract (according to comment #10) and we modified the title to be more accurate:

**'Sensitivity of fatigue reliability in wind turbine components: effects of design turbulence and the Wöhler exponent'**

10) Perhaps, you can rewrite the abstract. For me, it only became clear after having read the paper. It might help to shorten it and to highlight the main topic of the paper.

Shortened (from 357 to 297 words) and revised (to elaborate the context and relevance of the title of the paper better):

**'Fatigue reliability assessment of wind turbine components involves three major sources of uncertainty: material resistance, loads, and the mathematical models that connect the other two. Many studies focus on increasing accuracy in the assessment of fatigue loads, for example, by probabilistic modeling of the turbulence standard deviation of the wind.**

**The IEC standard suggests different distributions in Editions 3 and 4 as alternatives for the representative turbulence in its Normal Turbulence Model (NTM). There are debates on whether the suggested distributions provide conservative reliability levels, as the established design safety**

**factors are calibrated based on the representative turbulence approach. The current study addresses the debate by comparing annual reliability based on different scenarios of NTM using a probabilistic approach. More importantly, it elaborates on the relative importance of load assessment's accuracy in the context of fatigue reliability.**

**Using DTU 10MW reference wind turbine and the First Order Reliability Method (FORM), we study the changes in the level and the sensitivity of the annual reliability considering three main random inputs. We perform the study in the blade root flapwise and the tower base fore-aft load channels, considering different fatigue exponents in each channel.**

**The results show that integration over distributions of turbulence in each mean wind speed results in less conservative annual reliability levels than representative turbulence. The difference in the reliability levels varies according to turbulence distribution and the fatigue exponent. In the case of the tower base, the difference in reliability after 20 years can be up to 50\%.**

**However, the model and material uncertainty have much higher effects on the reliability levels compared to loads' uncertainty. Knowledge about such differences in the reliability levels due to the choice of turbulence distribution is especially important as it impacts the extent of lifetime extension through reliability re-assessments.'**

11) Fig. 1: "Turbulence standard deviation of the wind and…" is not self-explanatory. What is meant by "and…"?

The main intention was to point to all the variables in the wind condition that form the highest variability in the fatigue loads. However, we agree that this can be confusing. Thus, we omitted 'and…' and changed the caption to:

'Flowchart of the main random inputs (Xi) and the output (Y) considered in the present study for the fatigue reliability assessment.' to point out that in the current study, we only consider turbulence standard deviation to change in the load input.

12) L. 127: You state that you use the "DTU 10MW offshore turbine". However, you use its onshore version. That should be clarified.

To clarify, we omitted the word 'offshore' and added below sentence to the ending of section 2.1:

'In the current study, we use the onshore version of this turbine.'

13) It would be nice if you explain why you use 200 random seeds (and not more or less).

Below text added to section 2.2:

'We have used 200 seeds since the results of a previous study (Mozafari et al., 2023) show that the estimation of fatigue loads almost converges in this number of seeds, and the variations are negligible.'

14) Eq. 1 and in the following: Please, be consistent in your notation, e.g., $V_{hub}$ and $v_{hub}$

Changed. We capitalized all 'V' s.

15) Please, make clear that Eq. 2 and 3 are derived from equations given in IEC 61400-1, 2005.

We added below sentence to clarify:

'Equations (2) to Eq. (4) refer to the NTM model in (IEC 61400-1, 2005).'

16) Eq. 2 to 6: Please, keep the units consistent.

Resolved by addressing comment #17 and clarifying the unit for the coefficients.

17) Eq. 2: I think there is a mistake in the equation.

Thank you very much for noticing. We corrected equation 2 and would like to inform you that this has been a typo and the calculations and codes are based on the correct equation.

18) Eq. 3: I think that it is supposed to be $I_{ref}$ and not $TI_{ref}$.

Right. The typo is corrected now.

19) Eq. 3: Closing parenthesis is missing.

Corrected.

20) Perhaps, it would be good to state that $T \sim LN\ (\mu_T, \sigma_T)$ for Eq. (2) and (3) and $T \sim Wbl\ (k, C)$ for Eq. (5) and (6) and to give F(T) explicitly. Otherwise, the meaning of k and C are not fully clear.

Very nice comment. We clarified via below sentences:

'Equations (3) to Eq. (4) refer to the NTM model in (IEC 61400-1, 2005).'

+

'Equation (6) and Eq. (7) present the shape and scale parameters of the Weibull distribution (T ~Wbl (K, C)), which the fourth edition of the IEC standard (IEC 61400-1, 2019) suggests.'

In addition, we added two equations explicitly representing the CDF in each of the two distributions.

21) Eq. 4: Please, make clear that this equation does not give the 90th percentile for each wind speed, but is only a linear regression approximating it.

We added below explanation to emphasize:
One must note that Eq. (5) is not the exact calculation of the representative turbulence and is only a linear regression approximating it.

22) Eq. 5: Is this equation correct? I think $I_{ref}$ has to be removed.

True. Corrected. We have rechecked and confirm that this has been a typo and the calculations and codes are based on the correct equation.

23) Fig. 2: The horizontal axis is the turbulence (T), i.e., the standard deviation of the wind speed, in m/s?

Yes. Unit added in the horizontal axis.

24) Fig. 2: The vertical axis is $\ln(1-F(T))$, where $T \sim LN\ (\mu_T, \sigma_T)$ or $T \sim Wbl(k,C)$?

In fact, the logarithm shown here is not a natural logarithm. For more clarification on the information above, we added below sentence:

'In this plot, the horizontal axis shows turbulence (T) levels and vertical axis refers to log10(1 − F (T)), where T ~ LN (μT , σT ) and T ~ Wbl(K, C).'

In addition, we changed the labels in Fig. 2 for precise clarification.

25) Eq. 7: I think that it should be $S^{-m}$. Otherwise, the number of allowed cycles would increase for an increasing load if $m$ is positive (what it is according to your definition in the following).

Typo corrected. Thank you for noticing.

26) L. 227: $M_{xi}$ is only the flapwise bending moment if there is no blade pitching. I think that it is an acceptable simplification for this work, but it should be mentioned.

Below explanation added to clarify:

'In addition, in the current study, the pitch angle in the blade root is zero. Thus, the moments along the x and y axes refer to flapwise and edgewise moments, respectively.'

27) Eq. 11: Use $I_y$ and not $I$ to be consistent, and $(c/I_y)^m$ should not stand in the denominator on the right-hand side of the equation.

Done. The correction is also right and implemented. We apologize for the typos.

28) L. 255: You state that $SS=200$. However, you only use $SS=6$ in the entire paper. Hence, this should be changed here. Furthermore, it might help to refer to Section 3.1 at this point to clarify why 200 random seeds are used, but only $SS=6$.

Very good point.

ss=200 is mentioned because E(DEL_lifetime^m) was previously established the equations (before applying comment #3). According to our previous study (cited), we believe that 200 is high enough to get close to the expected (converged)value of this parameter. Now that according to comment #3, we are changing the expectation to realization (which seems clearer for a reader to follow), SS=6 is mentioned and below explanation has been added:

'In all current study cases, SS equals 6 bootstrapped from a database of 200 seeds (see Table 1).'

29) Eq. 17: I think that the operator in this equation is not self-explanatory to all readers.

Changed the notation to the integral over the area (area of the negative/zero limit state function values) for simplification and easier understanding.

30) Eq. 19 and 21 (and in the following paragraph): I think it should be $E$ (DEL$_\text{lifetime}{}^m$).

Resolved based on the changes according to comment #3.

31) Eq. 19 and 21 (and in the following paragraph): Use $I_y$ and not $I$ to be consistent.

We agree and did apply this change to all the relevant equations.

32) L. 339: What is meant by R=10?

Clarified with mentioning 'load ratio' in the text plus a reference is given.

'We consider a load ratio of R = 10 for the fatigue properties of the composite (see (Mikkelsen, 2020) for further information regarding the load ratio and the reason of the choice). '

We kept the details in the reference in order to keep the flow and intention of the text; the fatigue testing details are out of context.

33) L. 340: Without knowing Section 3.1, the reader will wonder why $\log(E(DEL_{lifetime m}))$, or $\log(DEL_{lifetime})$ is a distribution. Therefore, you should refer to the bootstrapping in Section 3.1 (cf. comment 28)

Agreed. We now clarify in below sentence in the text:

'To apply FORM analysis, first, we fit distributions to the estimations of $\log (DEL_{lifetime})$ obtained from bootstrapping and calculated via Eq. (16) based on 10-minute simulations.'

34) Fig. 3: Units for the horizontal axis are missing.

Unit added (m/s).

35) L. 386: The statement "Fig. reveals that […]" is only correct for $F(t)<0.9$.

Agree. We modify the text as below for clarification:

Fig. 3 reveals that the turbulence levels within the same probability are higher in case 2 (lognormal distribution) compared to case 3 (Weibull distribution) below 90% quantile. The trend changes above 90% quantile.

36) L. 406 and Fig. 4: $DEL_{bin}$ or $E(DEL_{bin m})$? If it is $DEL_{bin}$, the "different seeds" do not make any sense. If it is $E(DEL_{bin m})$ the units in Fig. 4 are incorrect.

We clarify by:

'Fig. 4 shows $DEL_{\text{bin}}$ values averaged over all seeds in each turbulence level (Eq. 12) in cases 2 and 3.'

However, we must emphasize that this is not E(DEL_bin^m) as we consider E (DEL_bin^m) to be the result of integration over all seeds and all turbulence levels.

37) L. 413: The statement "This observation reveals […]" is not correct. Figure 4 does not provide any information about the scatter in each bin, as it shows average values for each bin, i.e., $E(\text{DEL}_{\text{bin}^m})$. You probably mean that the variability is higher for varying turbulence values.

We confirm that your assumption is true. To make our intention clearer to the reader, we rephrased the statement and updated the paragraph as:

Another observation from Fig. 4 is the relatively faster decrease of the tower-base $DEL_{bin}$ as a function of turbulence compared to the blade. In other words, the difference between $DEL_{lifetime}$ obtained from a single high turbulence level and a single low level is relatively higher. This is partly because of the resonance in the tower base (see (Mozafari et al., 2023)) which enhances the effect of turbulence level on fatigue loads. Thus, we expect the integration over all turbulence levels (see Eq. (14)) to be more effective in decreasing variability DELbin and thus, $DEL_{lifetime}$ estimations in the case of tower base. The following includes …

38) L. 433: I agree with the statement "The lower variance […]". However, you should demonstrate that this is actually the case by running case 1 with 6*20=120 seeds instead of 6 seeds and show these results in the appendix. Otherwise, the comparison is not fair.

We have done that in another work. We added the reference in the text as below:

' (see (Mozafari et al., 2023) to see the effect of summation on the variability in case 1)'

39) L. 434: I do not agree with your second reason "The other reason is […]". If you consider different turbulence levels, you have low and high values. Hence, the variability should be higher compared to the case where you only use high turbulence levels.

rephrased to clarify:

'The integration over the whole range results in an expected value, which is more robust than a single value (90% quantile in this case).'

40) Fig. 5: Horizontal axis is "normalized DEL".

Corrected.

41) Fig. 5: Where do I see the "best distribution fits" that are mentioned in the caption of the figure?

This text was extra and a mistake. We omitted it from the caption.

42) Section 3.2: How are the best fitting distributions determined? Out of which distributions is the best fitting distribution chosen? How is the goodness of the fit judged?

We added an explanation as below:

'The trial of different distributions (GEV, lognormal, Normal, and Weibull in this case) with the maximum likelihood method finds the best distribution for fitting in the current study.'

43) Table 4: I think it would help if an equation for the GEV is given somewhere. Otherwise, it is not clear what "Par 1", "Par 2" and "Par 3" are. Even for the lognormal and Weibull it would help (cf. comment 20)

Agreed. Notations added in all the relevant tables.

44) L. 468: The statement "[...] that we get the same reliability level in the first year" only refers to case 1 if I am not mistaken.

True. We added below explanation to clarify:

'... a way that we get the same reliability level in the first year in case 1 to set a benchmark for comparisons.'

45) L. 510: Which DEL is meant here?

$DEL_{lifetime}$ (Text updated accordingly).

46) L. 523 and 554: Lifetime DEL? Or which one?

$DEL_{lifetime}$ (Text updated accordingly).

47) L. 524: "This has been made clearer in the sensitivity analysis": Where exactly has it been made clearer? I do not find this.

Modified the text as below to be clearer:

'This has been made clearer in the sensitivity analysis, where the effects of load variations are shown to be relatively negligible.'

48) L. 534: $I_{ref}$=0.1? Or what do you mean here?

Yes. We mean $Iref$ to be equal to 0.1. We added the term 'equal to 0.1' in the text to be clear.

49) L. 547: You state that MC can only be done when having the computational resources. However, you only need to evaluate Eq. (21), which should not be computationally very demanding, when knowing the distributions. Running the aero-elastic simulations probably takes much more time. Hence, the use of FORM instead of MC should be discussed in more detail. Or am I mistaken, and the procedure is computationally demand. In this case, please explain why this is the case.

For low probabilities of failure such as in structural components of the wind turbines, a lot of simulations are needed for MC. In a Monte Carlo analysis with N number of simulations, the coefficient of variation of the estimate ($P_f=10^8$ in our case) is proportional to $\frac{1}{\sqrt{N}}$ (based on the law of large numbers). This means that if the real probability of failure in a structural component is in the order of $10^{-x}$, approximately $10^{(x+2)}$ simulations are needed to achieve an estimate with a coefficient of variance in the order of 10%. Normal computers can save data with size up to N=$10^9$, meaning that we would be able to capture max probability of failure $10^{-7}$.It is possible to cluster the simulations for example to get 10 clusters of $10^9$ to be able to capture $P_f=10^8$. However, more and more loops are going to take a lot of time for normal processors.

50) L. 565: I would not call it "sensitivity analysis" but perhaps "importance ranking". You already stated in Section 3.3 that it is not really a representative sensitivity analysis.

Changed for the sake of accuracy in wording.

Further explanation: The only reason we claim that the values are not representative/real is because of calibrated (and not real) mean values for material properties in the study. However, the sensitivity analysis via FORM is in fact sorting the variables in order of sensitivity of the reliability to changes in each. The higher the importance the higher is this sensitivity. Overall, we change the word to 'importance ranking' to be more accurate but we would like to emphasize here that this is in fact a sensitivity analysis.

**Typos etc.:**

51) As you can see in the following, there are quite a lot of typos and inconsistencies. As I have definitely not found all of them, I recommend thorough proof reading.

52) Please, revise your citation style. It seems to be inconsistent.

53) L. 134: Remove the second parenthesis before "Larsen and Hansen, 2007".

54) Footnote 1: "in the time domain – developed in" not "in the time domain- developed in"

55) Eq. 1: $\exp(x)$ should be $e_x$ and pi should be $\pi$

56) Fig. 2: "Lognormal" and not "lognormal"

57) L. 199: $10_3$ and not 1e3.

58) L. 205 (and several times more): $k$ and not $K$

59) Eq. 9: Please, keep indices consistent, e.g., $I_y$ and not $Iy$

60) Table 2: "radius" not "radious"

61) L. 270: "Equation (15) shows" not "Equation (15 )shows".

62) L. 294: Missing citation "marquez2012fatigue".

63) L. 307: "resistance" not "Resistance"

64) L. 335 and 337: $10_{-4}$ and not e-4

65) Fig. 4: "Probability" not "Pobability" and 0-100% and not 0-20 (both horizontal axis).

66) Fig. 4: "MNm" not "Mnm" (vertical axis).

67) Fig. 4: 0 to 100% and not 0-20 (colour axis)

68) Table 4: "Par 2" and not "par2" and "Par 3" and not "par3"

69) L. 464: "Eq. (29) and" and not "Eq. (29) and"

70) L. 466: I think that there is something missing in the statement "the distributions in 3 are […]"

71) Fig. 6: "b) tower" and not "b)tower"

72) L. 470: "show that in both" and not "show that the in both"

73) Fig. 7: Please, update the legend, e.g., $\ln(\Delta)$ and not "log Delta" and which DEL is meant here?

74) L. 569: "the" and not "The"

Reply to all comments from #51 to #74 (including):

All applied and corrected in the text. Proofreading done.

---

## Author Comment (AC2)

**Sensitivity analysis of wind turbine fatigue reliability: effects of design turbulence and the Wöhler exponent (Manuscript number: wes-2023-47)**

Shadan Mozafari[1], Paul Veers[2], Jennifer Marie Rinker[1] , and Katherine Dykes[1]

1 Department of Wind Energy, Technical University of Denmark, Roskilde, Denmark
2 National Renewable Energy Laboratory (NREL), Golden, CO, USA.

**Response letter to referee #2 comments (Manuscript number: wes-2023-47):**

We would like to thank the reviewer for the positive feedback and for helping us improve the work by sharing important thoughts and comments. All comments are addressed, and a revised version of the article is prepared. In the following, all comments from reviewer #2 are mentioned, with the response from us after each.

**Comments and responses:**

*'This is a very well documented study, which is of value to the community and of particular practical relevance. My only comments refer to minor clarifications, as follows:'*

- Figure 1 is necessary, but somehow overly simplified. Consider enriching it to fully account for the procedure and the parameters/variables involved in the simulation.

This is a very fair comment. Thank you. Figure 1 is now updated and moved to the beginning of the 'methodology' section.

- In order to carry out a parametric analysis, the study has to be framed within a very specific context. It would be good to summarize in one position all of the simplifying assumptions made and how these may impact generalization of the outcomes (e.g. only blade flapwise and tower base fore-aft load channels considered - would considerations exist for further quantities? what is the effect of using FORM?).

Some additional results are added for the case of Siemens 2.3MW and The limitations for generalization of the results are added within a specific context to the 'Discussion section' as below points:

'

1. *Design class of the wind turbine: The thickness of the tail in the lognormal distribution is dependent on its standard deviation. The standard deviation of the distribution in different cases of NTM is a function of the reference turbulence level (see Eq. (3) to Eq. (8)). This means that there is a possibility that the results of the current study change with the wind turbine class. One case of lower reference turbulence intensity equal to 0.1 is tested in the current study showing the same trends. However, we encourage similar studies on different classes of wind turbines to track the possible differences in the trends and results. In addition, the annual mean wind speed is shown to influence the variability of the long-term fatigue loads and thus, overlap between different cases of the study. We suggest further studies on the lower mean wind speeds and the combination of the class with other changes.*

2. *Additional averaging of data in case of full distributions: A potential concern with the results is the difference in sample size for the different cases. There are more 10-minute simulations involved in estimations in cases 2 and 3. A larger sample size naturally decreases the variance in the $DEL_{lifetime}$ evaluations due to the law of large numbers (see (Mozafari et al., 2023) for more details). To investigate whether this is significant, we checked the effect by using different combinations of seed numbers, and the corresponding effect on the trends is negligible. However, we encourage testing of different calculation approaches to track any possible changes in the variability of fatigue loads in cases of using full distributions of the turbulence.*

3. *Other load cases: Among the standard design load cases related to fatigue, idling, and power production with fault also include the Normal Turbulence Model in the IEC standard. It is valuable to perform the same study considering these other load cases and their corresponding probabilities. In addition, considering all relevant load cases for fatigue (including shutdowns and start-ups) can change the long-term fatigue distributions and trends and should be considered in future studies.*

4. *Specificity of the wind turbine response: The main study uses the DTU 10MW wind turbine as the case study. The size and design of the wind turbine and its controller's design affect the turbine's response to a specific wind input. The Siemens 2.3 MW wind turbine (with a smaller size) but similar controller and class is checked, and the results show the same trends in distributions of the long-term fatigue load. In future studies, testing other wind turbines with a different type of controller and also using other aeroelastic simulation tools is beneficial.*

5. *Variability of the material properties and damage accumulation rule: The coefficients of variation: The variability of the initial material fatigue strength and Miner's rule are taken from the literature. Updating these two inputs can change the levels in the sensitivity analysis.*

6. *Variability of the fatigue loads: The only variable in defining the fatigue loads is the variation in the turbulence inputs; while this elaborates the shares of load uncertainty due to this specific variation, the sensitivity results can change when considering other sources of uncertainty on the load side.*

7. *Method of reliability assessment: The first-order reliability method performs well in very low probabilities of failure and less accurately in higher failure probabilities. Doing the same reliability analysis using Monte Carlo instead of FORM can provide more accurate reliability estimates if computational resources are available (see appendix for a detailed explanation regarding computational expenses of MC).*

8. *Offshore versus onshore: The study contains aeroelastic simulations with only wind inputs (onshore case study). However, in the case of offshore, the effects of wind turbulence on the structure response change, especially in the case of tower loads. Thus, we recommend performing the same study for offshore cases to investigate the possible changes in the trends.'*

- Moreover, how does this choice of a specific model (e DTU 10MW is an offshore wind turbine from IEC standard class 1A, IEC 61400-1, 2019) impact the generalization of results to further typical designs?

The change in the response of the wind turbine due to turbulence change depends on the controller. In addition, the turbulence class defines the benchmark scenario. These two aspects are emphasized again in the discussion now (points #1 and #4 above)

- How is the negligence of the start-up or shutdown events expected to impact the overall computation. Is a separate study dedicated to these short-term by high impact effects perhaps meaningful on this issue in the future?

This relevant point is added to discussion point #3 now. Thank you for mentioning.

- It is appreciated that further simulations are executed for sanity, e.g., " We re-performed the study for a lower reference turbulence intensity more suitable for offshore cases (0.1) and the trends were the same in terms of DEL distribution biases". However, these should be reported (e.g., in an Appendix).

This is a fair comment. The results for one example case (Tower base with m=3, all cases), are now added to appendix (Figure A1).

- For ensuring transparency and contributing toward open science, it would be meaningful (and highly valuable) for the authoring team to additionally publish the generated dataset and make it openly available.

This is a very good suggestion. The data is now uploaded to Gitlab with the link below:

Paper3 · main · Shadan Mozafari / PhD research papers · GitLab (dtu.dk)

However, the data for Siemens check case are not added to the above folder because of confidentiality.

- It would be helpful to comment in the Conclusions section, how the design of (simulated) experiments may influence the drawn conclusions and to elaborate on the aspects that could be refined/improved in future/further studies.

Thank you for your fair comment. This is now added to the conclusion ending. However, a more detailed explanation is added to the discussion section (mostly focused on point #2 in the discussion).

**Minor Comments**

- I very much appreciate the clear and concise writing style. Some (few) typos are present and it would be important to conduct a thorough proofreading for catching these and further ensuring a consistent notation for the symbols used in equations.

Proofreading is done again. Thank you.

- It would be good to ensure that brackets have a size that is appropriate to the enclosed quantity every time (e.g. for fraction, they should be bigger)

Applied and revised now.

- From an aesthetics point of view, it would be preferable for font types and sizes to agree with the caption style and size.

Thank you very much for your attention and valid comment. We fully agree with you. However, all the font and styles are based on the journal template setups.

---

## Author Response (AR1)

Response letters including changes are uploaded before (on Oct. 12$^{th}$)

---

## Referee Report (RR1)

**"Sensitivity of fatigue reliability in wind turbines: effects of design turbulence and the Wöhler exponent" (Manuscript number: wes-2023-47_R1)**

Thank you very much for the nice revision of the paper. I think that the quality of the paper has been improved significantly. I have just a few minor comments on your revision.

1) Eq. 11: The notation $Mx_i$ is a bit confusing as it could be understood as $M \times x_i$. $M_{x_i}$ would be much clearer but this is just a matter of taste.

2) Eq. 13: Is it actually $E[DEL_s^m]$ and not just $DEL_s^m$, as you talk about a single sample of a 10-minute time series here? $E[DEL_s^m]$ would mean that you average something, but the averaging of several seeds follows in Eq. 14. Hence, I think it should be $DEL_s^m$.

3) Eq. 14: Is it actually $E[DEL_{bin}^m]$ and not just $DEL_{bin}^m$ on the left side of the equation? As the right side of the equation means to take the mean value, the current definitions would mean $E[DEL_{bin}^m] = E[DEL_s^m]$, and therefore, $DEL_{bin}^m = DEL_s^m$. Furthermore, in Fig. 4, you show $DEL_{bin}$ (as written in the caption). And what you actually show is $(E[DEL_s^m])^{1/m} = DEL_{bin}$, is you rewrite Eq. 14 as follows: $DEL_{bin}^m = \sum_{s=1}^{SS} \frac{(DEL_s)^m}{SS}$. If you do this, you have to use $DEL_{bin}^m$ instead of $E[DEL_{bin}^m]$ in Eq. 15 and 16 as well. This would remove all $E[\dots]$ in the paper, which makes things much clearer.

4) L. 552-554: Do you actually mean Fig. B1 and not Fig. 9? Is it $m = 12$ and not $m = 10$ (in line 554), as Fig. 9 shows $m = 10$. If you mean Fig. 9, a reference to Fig. B1 is missing.

5) Fig. 11: A legend is missing.

6) Comment 42 of the first review: "Section 3.2: How are the best fitting distributions determined? Out of which distributions is the best fitting distribution chosen? How is the goodness of the fit judged?" You answered the first two questions, i.e., how did you fit (maximum likelihood estimations (MLE)) and which distributions (GEV etc.). However, you did not answer, how the best distribution is chosen (I do not mean the best distribution parameters, this is done using MLE, but actually the best distribution, i.e., GEV etc.).

7) Comment 49 of the first review: "L. 547: You state that MC can only be done when having the computational resources. […]" I am still not really convinced that MC is not suitable here. Even when using 1000 loops with 1 million evaluations of Eq. (21) each, the processor time is probably relatively small compared to processor time of the approximate 100,000 aero-elastic simulations and probably even low compared to using 6 seeds per bin (i.e., more than 1,000 aero-elastic simulations). However, it is fine to use FORM, as you showed that it is a sufficient approximation. Hence, you do not have to give further explanations on this topic in the revision.

Typos etc.:

8) Your notation is still not completely consistent, e.g., in Eq. 13, you write $Neq$ but in line 256 it is $N_{eq}$. Similar, in line 249, it is $M_i$ and in Eq. 12 it is $Mx_i$.

9) Table 5: Par 2 and Par 3.

10) Check you references, as, for example, Sørensen is not written correctly (l. 767).

---

## Author Response (AR2)

**Response letter - "Sensitivity of fatigue reliability in wind turbines: effects of design turbulence and the Wöhler exponent" Manuscript number: wes-2023-47**

**Response to referee #1:**

We would like to thank the reviewer 1 for the final check and comments which helped us improve the final revision of the paper. The comments from the reviewer and their responses are presented in the continuing with two colors of black and blue, correspondingly.

Thank you very much for the nice revision of the paper. I think that the quality of the paper has been improved significantly. I have just a few minor comments on your revision.

1) Eq. 11: The notation $Mxi$ is a bit confusing as it could be understood as $M \times xi$. $M_{xi}$ would be much clearer but this is just a matter of taste.

   Thank you. Noted and corrected.

2) Eq. 13: Is it actually $E[DELs^m]$ and not just $DELs^m$, as you talk about a single sample of a 10-minute time series here? $E[DELs^m]$ would mean that you average something, but the averaging of several seeds follows in Eq. 14. Hence, I think it should be $DELs^m$.

   Corrected.

3) Eq. 14: Is it actually $E[DEL_{bin}^m]$ and not just $DEL_{bin}^m$ on the left side of the equation? As the right side of the equation means to take the mean value, the current definitions would mean $E[DEL_{bin}^m]=E[DELs^m]$, and therefore, $DEL_{bin}^m=DEL_s^m$. Furthermore, in Fig. 4, you show $DEL_{bin}$ (as written in the caption). And what you actually show is $(E[DEL_s^m])^{1/m}=DEL_{bin}$, is you rewrite Eq. 14 as follows: $DEL_{bin}^m=\sum_{s=1}^{SS}\frac{(DELs)^m}{SS}$. If you do this, you have to use $DEL_{bin}^m$ instead of $E[DEL_{bin}^m]$ in Eq. 15 and 16 as well. This would remove all $E[...]$ in the paper, which makes things much clearer.

   That is true since all the notations are realizations of DEL_bin. Corrected and changed to $DELbin$ in equations 14, 15, and 16.

4) L. 552-554: Do you actually mean Fig. B1 and not Fig. 9? Is it $m=12$ and not $m=10$ (in line 554), as Fig. 9 shows $m=10$. If you mean Fig. 9, a reference to Fig. B1 is missing.

   Corrected. Thank you for mentioning.

5) Fig. 11: A legend is missing.

   True. Corrected.

6) Comment 42 of the first review: "Section 3.2: How are the best fitting distributions determined? Out of which distributions is the best fitting distribution chosen? How is the goodness of the fit judged?" You answered the first two questions, i.e., how did you fit (maximum likelihood estimations (MLE)) and which distributions (GEV etc.). However, you did not answer, how the best distribution is chosen (I do not mean the best distribution parameters, this is done using MLE, but actually the best distribution, i.e., GEV etc.).

Completed the sentence in the section 3.2 to: 'We find the best distribution fits among different options (GEV, lognormal, normal, and Weibull in this case) using maximum likelihood method and Akaike information criterion (Akaike, H., 1973).' The corresponding reference is also added.

7) Comment 49 of the first review: "L. 547: You state that MC can only be done when having the computational resources. […]" I am still not really convinced that MC is not suitable here. Even when using 1000 loops with 1 million evaluations of Eq. (21) each, the processor time is probably relatively small compared to processor time of the approximate 100,000 aero-elastic simulations and probably even low compared to using 6 seeds per bin (i.e., more than 1,000 aero-elastic simulations). However, it is fine to use FORM, as you showed that it is a sufficient approximation. Hence, you do not have to give further explanations on this topic in the revision.

We appreciate your comment. The simulations for the study are performed using HPC cluster. Thus, we agree with your statement: if such facility is not accessible, the whole study would be computationally more expensive and if available, the Monte Carlo can also be done using the same. However, the reliability analysis was done using personal computer and here the comparison in is between FORM and MC not their computational expense compared to the aeroelastic simulations.

Typos etc.:
8) Your notation is still not completely consistent, e.g., in Eq. 13, you write $Neq$ but in line 256 it is $Neq$. Similar, in line 249, it is $Mi$ and in Eq. 12 it is $Mxi$.

Thank you for noting. Corrected in the text.

9) Table 5: Par 2 and Par 3.

Corrected.

10) Check you references, as, for example, Sørensen is not written correctly (l. 767).

Corrected.

**Response to referee #2:**

We would like to thank reviewer #2 for the nice feedback. We are glad that the final version met the reviewer's expectations.